# Pleiotropic effects of BAFF on the senescence-associated secretome and growth arrest

Martina Rossi[1], Carlos Anerillas[1], Maria Laura Idda[2], Rachel Munk[1], Chang Hoon Shin[1], Stefano Donega[1,3], Dimitrios Tsitsipatis[1], Allison B Herman[1], Jennifer L Martindale[1], Xiaoling Yang[1], Yulan Piao[1], Krystyna Mazan-Mamczarz[1], Jinshui Fan[1], Luigi Ferrucci[3], Peter F Johnson[4], Supriyo De[1], Kotb Abdelmohsen[1]*, Myriam Gorospe[1]*

[1]Laboratory of Genetics and Genomics, National Institute on Aging (NIA) Intramural Research Program (IRP), National Institutes of Health, Baltimore, United States; [2]Institute for Genetic and Biomedical Research (IRGB), National Research Council, Sassary, Italy; [3]Translational Gerontology Branch, NIA IRP, NIH, Baltimore, United States; [4]Mouse Cancer Genetics Program, Center for Cancer Research, National Cancer Institute IRP, Frederick, United States

*For correspondence:
abdelmohsenk@grc.nia.nih.gov
(KA);
myriam-gorospe@nih.gov (MG)

**Competing interest:** The authors declare that no competing interests exist.

**Abstract** Senescent cells release a variety of cytokines, proteases, and growth factors collectively known as the senescence-associated secretory phenotype (SASP). Sustained SASP contributes to a pattern of chronic inflammation associated with aging and implicated in many age-related diseases. Here, we investigated the expression and function of the immunomodulatory cytokine BAFF (B-cell activating factor; encoded by the *TNFSF13B* gene), a SASP protein, in multiple senescence models. We first characterized BAFF production across different senescence paradigms, including senescent human diploid fibroblasts (WI-38, IMR-90) and monocytic leukemia cells (THP-1), and tissues of mice induced to undergo senescence. We then identified IRF1 (interferon regulatory factor 1) as a transcription factor required for promoting *TNFSF13B* mRNA transcription in senescence. We discovered that suppressing BAFF production decreased the senescent phenotype of both fibroblasts and monocyte-like cells, reducing IL6 secretion and SA-β-Gal staining. Importantly, however, the influence of BAFF on the senescence program was cell type-specific: in monocytes, BAFF promoted the early activation of NF-κB and general SASP secretion, while in fibroblasts, BAFF contributed to the production and function of TP53 (p53). We propose that BAFF is elevated across senescence models and is a potential target for senotherapy.

## Editor's evaluation

Rossi et al. carry out a valuable characterization of the molecular circuitry connecting the immunomodulatory cytokine BAFF (B-cell activating factor) in the context of cellular senescence. They present solid evidence that BAFF is upregulated in response to senescence, and that this upregulation is partially driven by the immune response-regulating transcription factor (TF) IRF1, with potential cell type-specific effects during senescence. Ultimately, these results strongly suggest that BAFF plays a senomorphic role in senescence, modulating downstream senescence-associated phenotypes, and may be an interesting candidate for senomorphic therapy.

## Introduction

Cellular senescence is a state of indefinite cell cycle arrest arising in response to a variety of stressful stimuli, including telomere erosion, oncogenic signaling, and damage to DNA and other molecules (*Hayflick, 1965*; *Muñoz-Espín and Serrano, 2014*). Despite experiencing persistent growth arrest, senescent cells remain metabolically active and express and secrete distinct subsets of proteins, including cytokines, chemokines, metalloproteases, and growth factors, a trait collectively known as the senescence-associated secretory phenotype (SASP). The sustained production of SASP factors in tissues and organs promotes the recruitment of immune cells, tissue remodeling, and chronic inflammation at the systemic level (*Basisty et al., 2020*; *Franceschi and Campisi, 2014*). Senescent cells are necessary for developmental processes like embryonic morphogenesis and wound healing, and have tumor-suppressive properties in young individuals (*Campisi, 2013*; *He and Sharpless, 2017*). However, with advancing age, the accumulation of senescent cells exacerbates age-related pathologies like cancer, diabetes, and neurodegenerative and cardiovascular diseases (*Franceschi et al., 2007*; *McHugh and Gil, 2018*; *Muñoz-Espín and Serrano, 2014*).

Despite the recognized impact of senescent cells, the unequivocal detection of senescent cells remains a challenge. Several markers of senescence have been described in cultured cells as well as in tissues and organs, but they are not universal markers of all senescent cells, and they are often not exclusive of senescent cells, as non-senescent cells may express them as well. Therefore, multiple markers must be present in order to identify a cell as senescent. Classic markers of senescence include cell cycle inhibitors (e.g. the cyclin-dependent kinase [CDK] inhibitor proteins p16 [CDKN2A] and p21 [CDKN1A]), the presence of nuclear foci of unresolved DNA damage (often visualized by the presence of a phosphorylated histone, γ-H2AX), and increased activity of a senescence-associated β-Galactosidase (SA-β-Gal) that functions at pH 6 (*Gorgoulis et al., 2019*; *Hernandez-Segura et al., 2018*; *Idda et al., 2020*). Our recent screen identified the *TNFSF13B* (*Tumor Necrosis Factor Superfamily Member 13B*) mRNA, encoding the cytokine BAFF (B cell-activating factor), as an RNA marker shared across a number of senescent cell types and inducers (*Casella et al., 2019*). BAFF is produced as a membrane-bound or secreted cytokine that plays an essential role in the homeostasis of the immune system (*Kalled, 2005*; *Mackay and Schneider, 2009*; *Rauch et al., 2009*; *Smulski and Eibel, 2018*). However, BAFF also plays a key role in sustaining the inflammatory processes associated with autoimmune diseases like systemic lupus erythematosus, multiple sclerosis, and rheumatoid arthritis (*Idda et al., 2019*; *Kalled, 2005*; *Steri et al., 2017*). The pro-inflammatory role of BAFF is primarily elicited by the activation of three receptors (BAFFR [TNFRSF13C], TACI [TNFRSF13B], and BCMA [TNFRSF17]), which converge on paths that signal through the transcription factor NF-κB (*Bossen and Schneider, 2006*; *Eslami and Schneider, 2021*; *Nicoletti et al., 2016*; *Smulski and Eibel, 2018*). In turn, NF-κB transcriptionally activates the production of many pro-inflammatory and adhesion factors (*Tang et al., 2018*). However, the role of BAFF in cell senescence is unknown.

Here, we investigated the expression and function of BAFF in senescence. We present evidence that BAFF is elevated in models of cell senescence in mice and cultured human cells. In response to a range of inducers, the levels of *TNFSF13B* mRNA and total cellular BAFF were increased, as were the levels of secreted BAFF in the culture media of senescent cells and in the blood of mice following doxorubicin-induced senescence. Mechanistically, the transcription factor IRF1 (interferon regulatory factor 1) was found to increase *TNFSF13B* mRNA levels in senescent cells via activated transcription. In the presence of secreted BAFF, the human monocytic leukemia-derived, p53-deficient cell line THP-1 activated NF-κB, which in turn transcriptionally induced the production of SASP factors, while p53-proficient human senescent fibroblasts activated a p53-dependent gene expression program. Interestingly, in senescent p53-proficient mouse macrophage/monocyte-like RAW 264.7 cells, BAFF promoted both p53 phosphorylation and SASP. Our data indicate that BAFF promotes senescence in a pleiotropic manner, enhancing the SASP in some cell types and p53-mediated growth arrest in others.

## Results

### BAFF increases in cultured senescent cells and in a senescent mouse model

Our previous analysis of multiple models of senescence, including human primary fibroblasts (WI-38, IMR-90), umbilical vein endothelial cells (HUVECs), and aortic endothelial cells (HAECs), revealed a unique senescence transcriptome signature (*Casella et al., 2019*), including heightened production of the *TNFSF13B* mRNA, encoding the cytokine BAFF, a SASP factor. RNA sequencing (RNA-seq) analysis (*Casella et al., 2019*) indicated that the levels of *TNFSF13B/BAFF* mRNA were elevated across all the senescence models tested previously. These models included WI-38 fibroblasts that were proliferating at population doubling level (PDL) 25 and then rendered replicatively senescent (RS) by division until exhaustion (~PDL55), exposed to ionizing radiation (IR at 10 Gy, evaluated 10 days later), infected for 18 hr with a lentivirus that triggered oncogene-induced senescence (OIS) by expression of HRAS$^{G12V}$ and evaluated 8 days later, or treated with doxorubicin (DOX) for 24 hr and evaluated 7 days later. Additional models tested included proliferating IMR-90 fibroblasts (PDL25) rendered RS by culture to ~PDL55 or senescent by exposure to IR (10 Gy, assessed 10 days later), and HUVECs and HAECs which were either proliferating or rendered senescent by exposure to IR (4 Gy, evaluated 10 days later) (*Figure 1—figure supplement 1A*).

Given that BAFF is mainly expressed by myeloid cells like monocytes and dendritic cells (*Sakai and Akkoyunlu, 2017*; *Steri et al., 2017*), we extended the analysis of BAFF expression in senescence using the human monocytic leukemia-derived cell line THP-1. First, we induced senescence in THP-1 cells by treatment with 5 Gy IR, a dose selected because it suppressed cell growth but did not reduce cell viability (*Figure 1—figure supplement 1B and C*). Six days after exposure to IR, the rate of proliferation declined, SPiDER-β-Gal staining was elevated, and the proportion of G2/M cells increased modestly relative to untreated cells (*Figure 1—figure supplement 1D–F*).

Similarly, we selected 10 nM DOX treatment for 48 hr, followed by 4 additional days in DOX-free media, as a treatment dose that induced senescence in THP-1 cells without reducing viability (not shown). As additional models of senescence, we included primary WI-38 fibroblasts rendered senescent by exposure to IR or to HRAS$^{G12V}$ (OIS) as described above (*Figure 1A–D*). In these treatment groups, we first found markedly increased senescence-associated (SA)-β-Gal levels in all four senescence groups (*Figure 1A*). We then quantified the levels of *TNFSF13B* mRNA by reverse transcription (RT) followed by real-time, quantitative (q)PCR analysis and normalized them to the levels of *ACTB* mRNA, encoding the housekeeping protein ACTB (β-Actin) (*Figure 1B*); as shown, *TNFSF13B* mRNA levels increased markedly in all four senescence groups.

We subsequently measured BAFF protein levels by western blot analysis alongside senescence protein markers DPP4 and p21, known to increase during senescence, and loading control ACTB (*Figure 1C*). Finally, given that BAFF exerts its function as a membrane-bound protein and a secreted cytokine (*Eslami and Schneider, 2021*), we measured BAFF in conditioned media using ELISA; as shown (*Figure 1D*), the levels of secreted BAFF increased in conditioned media from senescent cells. The levels of *TNFSF13B* mRNA and secreted BAFF were also measured in other models of senescence, including IMR-90 fibroblasts treated with etoposide (ETO, 50 μM for 72 hr followed by 7 additional days without ETO), WI-38 fibroblasts rendered senescent by RS or ETO, and human vascular smooth muscle cells (hVSMCs) exposed to 10 Gy IR and cultured for an additional 7 days, where enhanced SA-β-Gal activity was confirmed (*Figure 1—figure supplement 1G*). In all cases, *TNFSF13B* mRNA levels increased during senescence, and secreted BAFF was generally elevated with senescence, although it was undetectable in hVSMCs (*Figure 1—figure supplement 1H, I*).

Next, we evaluated the levels of BAFF in mice in which senescent cells were induced to accumulate in tissues and organs. We triggered a rise in senescent cells in mice by injecting DOX (10 mg/kg) once, and measuring BAFF levels in the serum 14 days later (*Figure 1E* and Materials and methods). We observed a modest but significant increase in the levels of circulating BAFF (*Figure 1F*), with a difference greater than 1600 pg/ml in the median between the two groups. As a positive control that senescence was induced in the mouse, we measured increased circulating levels of the core SASP factor GDF15 (*Figure 1F*; *Basisty et al., 2020*). Next, we analyzed the expression of BAFF in the spleen, a major reservoir of myeloid cells. We observed a significant increase in the levels of *Tnfsf13b* mRNA, as measured by RT-qPCR analysis; we measured the levels of senescence marker *Cdkn1a* (*p21*) mRNA alongside as a positive control (*Figure 1G*). To study if the elevation in *Tnfsf13b* mRNA levels yielded

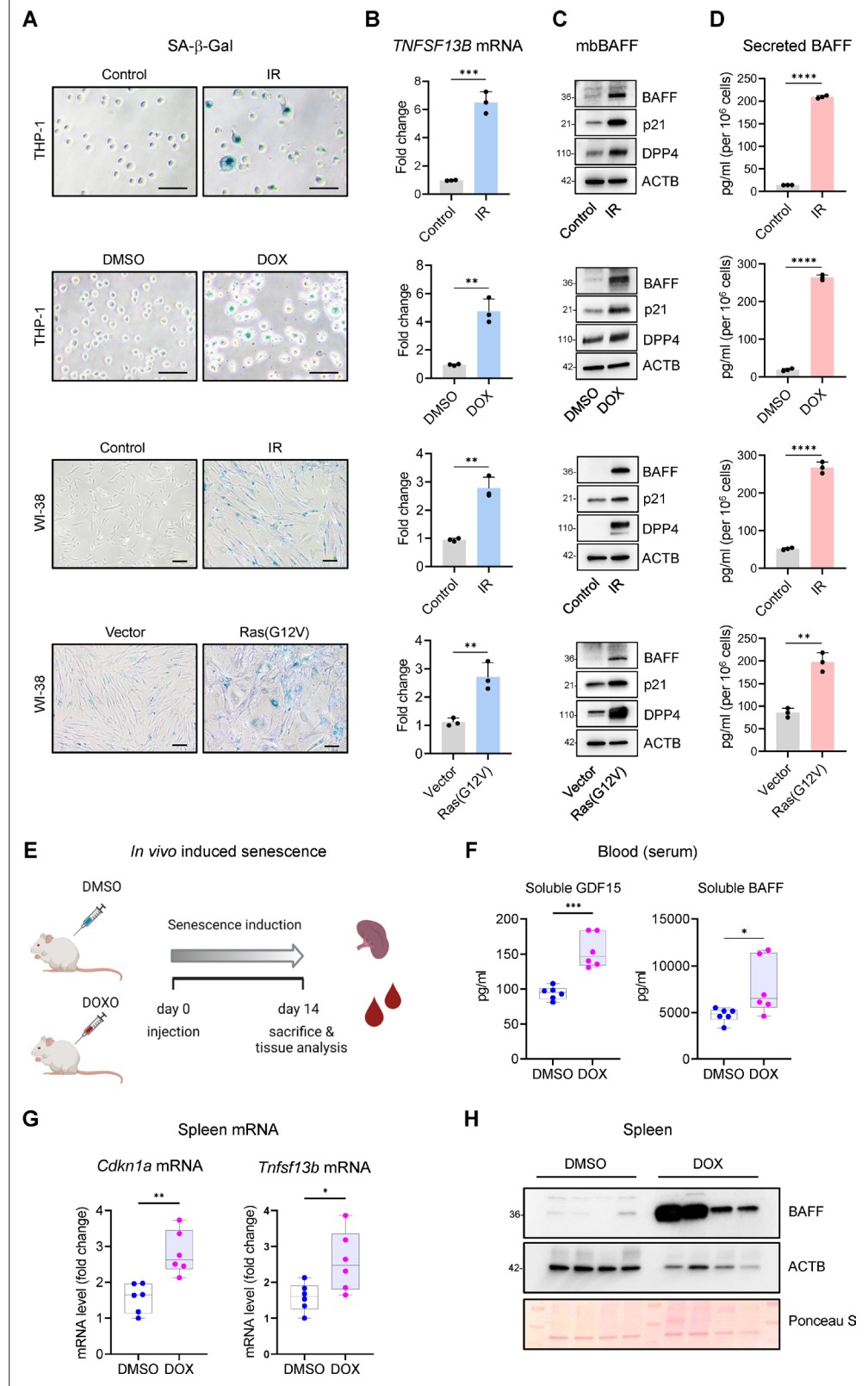

**Figure 1.** Increased BAFF expression in cultured senescent cells and in a mouse model of senescence. (**A**) Micrographs of SA-β-Gal activity in senescent cultured cells relative to corresponding proliferating controls. From top: THP-1 cells untreated (Control, proliferating) or treated with a single dose of IR (5 Gy) and cultured for an additional 6 days; THP-1 cells treated with DMSO or with a single dose of 10 nM doxorubicin (DOX) for

*Figure 1 continued on next page*

*Figure 1 continued*

48 hr, followed by 4 additional days in culture without DOX (for a total of 6 days since the treatment with DOX); WI-38 fibroblasts that were proliferating or had been treated with a single dose of IR (10 Gy) and cultured for an additional 10 days; WI-38 fibroblasts transduced for 18 hr with a control lentivirus (empty vector) or with a RAS(G12V) lentivirus, whereupon culture medium was replaced and cells were cultured for an additional 8 days. (**B**) RT-qPCR analysis of the levels of *TNFSF13B* mRNA, normalized to the levels of *ACTB* mRNA (encoding the housekeeping protein ACTB [β-Actin]) in cells processed as in (**A**). (**C**) Western blot analysis of the levels of membrane-bound BAFF (mbBAFF), DPP4, p21, and loading control ACTB in cells processed as in (**A**). (**D**) Levels of BAFF secreted in the culture media in cells treated as described in (**A**), quantified by ELISA. (**E**) Schematic of the strategy to induce senescence in DOX-treated mice (Materials and methods), created with BioRender. (**F**) Quantification of the levels of soluble GDF15 and BAFF in the blood (serum) of mice treated as in (**E**) using ELISA (Materials and methods). GDF15 was used as a positive control of induced senescence (***Basisty et al., 2020***). (**G**) RT-qPCR analysis of the levels of *Cdkn1a* and *Tnfsf13b* mRNAs in spleens of mice treated as in (**E**). *18* S rRNA levels were measured and used for data normalization. (**H**) Representative western blot analysis of the levels of BAFF in spleen homogenates obtained from mice treated as in (**E**). ACTB and Ponceau S staining served as loading controls. Significance (\*, p<0.05; \*\*, p<0.01; \*\*\*, p<0.001; \*\*\*\*, p<0.0001) was assessed by Student's *t*-test in all panels. Scale bars, 100 μm.

The online version of this article includes the following source data and figure supplement(s) for figure 1:

**Source data 1.** Uncropped immunoblots for *Figure 1*.

**Figure supplement 1.** Extended data from *Figure 1*.

---

increased BAFF protein production, we performed western blot analysis on homogenates from spleen and observed high levels of BAFF protein in DOX-treated mice compared to control (DMSO-treated) mice (*Figure 1H*). Given the high variability in ACTB expression among different mice, we also used Ponceau S staining to monitor loading differences in western blots (*Figure 1H*). Together, these results indicate that BAFF is elevated in senescent cultured cells and also in senescent cells in vivo.

## IRF1 promotes *TNFSF13B* mRNA transcription in senescence

Next, to study the mechanism(s) underlying BAFF production in senescence, we investigated if the rise in *TNFSF13B* mRNA in irradiated THP-1 cells was the result of transcriptional or posttranscriptional regulatory processes. In THP-1 cells rendered senescent by exposure to IR, we assessed the changes in levels of *TNFSF13B* pre-mRNA (a surrogate measure of de novo transcription) by RT-qPCR analysis and found that they mirrored the levels of total *TNFSF13B* mRNA (*Figure 2A*). These results indicated that the rise in *TNFSF13B* mRNA was strongly driven by increased transcription. The notion that the rise in *TNFSF13B* mRNA levels was the result of increased transcription and not increased *TNFSF13B* mRNA stability was directly tested by measuring the half-lives of *TNFSF13B* mRNA after adding the transcriptional inhibitor Actinomycin D; as shown in *Figure 2—figure supplement 1A*, the rate of *TNFSF13B* mRNA decay in proliferating THP-1 cells ($t_{1/2}$~4 hr) was comparable to that observed in THP-1 cells rendered senescent by IR ($t_{1/2}$~3 hr), indicating that *TNFSF13B* mRNA was not longer-lived in senescent THP-1 cells, and further supporting the idea that *TNFSF13B* mRNA increased through elevated transcription. Similarly, WI-38 cells rendered senescent by RS or IR showed increased *TNFSF13B* pre-mRNA levels that mirrored the rise in *TNFSF13B* mRNA; *TNFSF13B* mRNA had comparable half-lives across the proliferating and senescent populations (*Figure 2—figure supplement 1B–D*), suggesting that *TNFSF13B* mRNA was transcriptionally elevated across senescence models. In mRNA stability assays, the unstable *MYC* mRNA was measured as an internal control (*Figure 2—figure supplement 1A, C and D*).

To begin to identify the transcription factors (TFs) that might upregulate *TNFSF13B* mRNA transcription in senescence, we analyzed subsets of proteins identified as changing, based on proteomic analysis, in THP-1 cells primed for senescence after irradiation with 5 Gy and cultured for 72 hr (*Figure 2—source data 1*). The proteomic analysis (*Figure 2B* and *Figure 2—source data 1*) showed a robust and predominant activation of the Type I interferon response pathway in IR-treated THP-1 cells. Importantly, DNA damage was previously shown to prime both the interferon response and inflammation (***Härtlova et al., 2015***; ***Li and Chen, 2018***). Computational analysis of the evolutionarily conserved regions (ECRs) in the *TNFSF13B* promoter and the rVista2.0 database (https://ecrbrowser. dcode.org/) indicated the presence of an interferon-sensitive response element (ISRE) and multiple

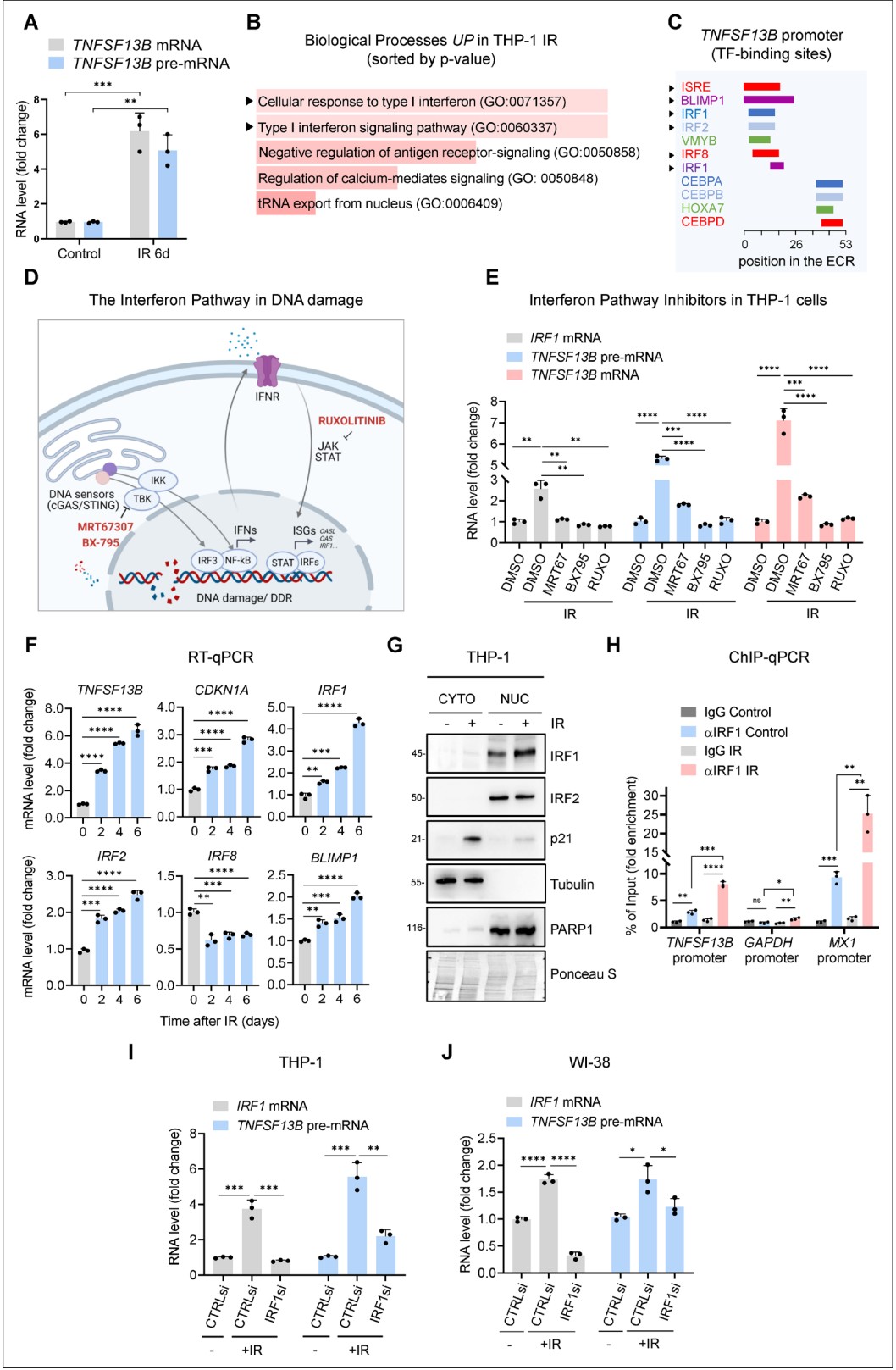

**Figure 2.** Interferon response pathway and IRF1 promote *TNFSF13B* transcription in senescence. (**A**) RT-qPCR analysis of the levels of *TNFSF13B* mRNA and *TNFSF13B* pre-mRNA in THP1 cells that were either untreated (Control) or treated with IR (5 Gy) and analyzed 6 days later. The levels of *ACTB* mRNA, encoding the housekeeping protein ACTB (β-Actin), were measured and used for data normalization. (**B**) Gene Ontology terms

*Figure 2 continued on next page*

*Figure 2 continued*

enriched after proteomic analysis of THP-1 cells that were rendered senescent by IR relative to untreated control cells. Proteomic analysis is available as *Figure 2—source data 1*. GO terms were sorted by p-value ranking (EnrichR analysis). (**C**) Schematic of conserved TF-binding sites on the *TNFSF13B* promoter, as analyzed using ECR browser and rVista 2.0. (**D**) Schematic of the interferon response triggered by DNA damage, highlighting the targets affected by the different interferon inhibitors used in our study (*Fu et al., 2020*; *Härtlova et al., 2015*; *Li and Chen, 2018*), created using BioRender. (**E**) RT-qPCR analysis of the levels of *TNFSF13B* mRNA, *TNFSF13B* pre-mRNA, and *IRF1* mRNA in THP-1 cells after treatment with the inhibitors of the interferon pathway indicated in (**D**) [Ruxolitinib (Ruxo, 1 μM), MRT67307 (MRT67, 5 μM), and BX-795 (5 μM)] or with control vehicle DMSO for 5 days after treatment with IR. *IRF1* mRNA was included as a positive control of interferon-stimulated mRNAs. *ACTB* mRNA levels were measured and used for data normalization. (**F**) RT-qPCR analysis of the expression levels of *TNFSF13B* and *p21* mRNAs and other interferon-regulated transcripts (*IRF1*, *IRF2*, *IRF8*, and *BLIMP1* mRNAs) in THP-1 cells that were left untreated or were treated with IR and cultured for the indicated times. *ACTB* mRNA levels were used for data normalization. (**G**) Western blot analysis of the levels of IRF1 and IRF2 in the cytoplasmic and nuclear fractions of THP-1 cells that were left untreated or were irradiated and assayed 6 days later. The cytoplasmic protein tubulin, the nuclear protein PARP1, and the senescence-associated protein p21 were included in the analysis. Ponceau staining of the transfer membrane was included to monitor differences in loading and transfer among samples. (**H**) ChIP-qPCR analysis of endogenous IRF1 in control (proliferating) or IR-treated THP-1 cells (5 Gy IR, assayed 72 hr later). The purified DNA was analyzed by qPCR using primers binding to *TNFSF13B* promoter, *GAPDH* promoter (negative control) and *MX1* promoter (positive control). Data are presented as fold enrichment of the antibody signal versus the negative control IgG using the comparative $2^{-\Delta\Delta Ct}$ method normalized to the percentage of the input. (**I**) RT-qPCR analysis of the levels of *IRF1* mRNA and *TNFSF13B* pre-mRNA in THP-1 cells transfected with control (CTRLsi) or IRF1-directed (IRF1si) siRNAs 72 hr after either no treatment (-) or treatment with IR. *ACTB* mRNA levels were used for data normalization. (**J**) RT-qPCR analysis of the levels of *TNFSF13B* pre-mRNA in WI-38 cells transfected with CTRLsi or IRF1si and either left untreated (-) or treated with IR (10 Gy) and assayed 5 days later. *ACTB* mRNA levels were measured and used for data normalization. Significance (ns, not significant; *, p<0.05; **, p<0.01; ***, p<0.001; ****, p<0.0001) was assessed by Student's *t*-test.

The online version of this article includes the following source data and figure supplement(s) for figure 2:

**Source data 1.** Proteomic analysis performed in control THP-1 cells treated with IR.

**Source data 2.** Uncropped western blot images for *Figure 2*.

**Figure supplement 1.** Extended Data from *Figure 2*.

**Figure supplement 1—source data 1.** Uncropped western blots.

---

binding sites for interferon regulatory factors (IRFs), including IRF1, IRF2, and IRF8 (*Figure 2C*). To test the possible role of these TFs in driving *TNFSF13B* transcription, we evaluated three inhibitors (Ruxolitinib [Ruxo], MRT67307 [MRT67], and BX-795) that target the interferon response pathway at different levels (*Figure 2D*), and measured the efficiency of these inhibitors by quantifying the levels of *IRF1* mRNA, a transcript that is inducible during the IFN response (*Figure 2D*; *Forero et al., 2019*; *Fujita et al., 1989*; *Panda et al., 2019*). As shown, these inhibitors decreased the levels of *TNFSF13B* pre-mRNA and *TNFSF13B* mRNA in IR-induced senescent THP-1 cells; they also decreased *IRF1* mRNA levels, included here as a positive control (*Figure 2E*). These results uncovered a key role for the interferon response in promoting BAFF expression following senescence-inducing DNA damage and strengthened the hypothesis that IRFs enhanced BAFF transcription.

To narrow down our list of IRF candidates possibly involved in BAFF transcription (*Figure 2C*), we focused on those IRF factors whose expression levels either increased or remained unchanged (but did not decline) after senescence-inducing DNA damage. RT-qPCR analysis at the times shown following treatment of THP-1 cells with IR (5 Gy) revealed increased levels of *TNFSF13B* and *CDKN1A* mRNAs, as well as increased levels of *IRF1*, *IRF2*, *BLIMP1* mRNAs and reduced *IRF8* mRNA levels (*Figure 2F*). Given that *IRF8* mRNA levels were strongly reduced in senescence and that BLIMP1 is a known transcriptional repressor (*Keller and Maniatis, 1991*; *Martins and Calame, 2008*), we focused on IRF1 and IRF2 as potential inducers of *TNFSF13B* mRNA transcription in senescent cells. THP-1 cell fractionation followed by analysis of changes in their subcellular distribution during senescence revealed elevated nuclear localization of IRF1, but not IRF2, in senescent cells (*Figure 2G*), suggesting that IRF1 might be primarily implicated in the transcription of *TNFSF13B* mRNA. Importantly, chromatin immunoprecipitation (ChIP) followed by qPCR analysis revealed increased IRF1 binding to the

*TNFSF13B* promoter in THP-1 cells after IR. We amplified the *GAPDH* and *MX1* promoters as negative and positive controls for IRF1 binding, respectively (*Panda et al., 2019*; *Figure 2H*).

Finally, to confirm the role of IRF1 in BAFF transcription upon IR, we silenced IRF1 in THP-1 cells using specific siRNAs. RT-qPCR analysis confirmed the silencing of IRF1 and showed a significant reduction in *TNFSF13B* pre-mRNA levels (*Figure 2I* and *Figure 2—figure supplement 1E*). Similarly, the levels of *IRF1* mRNA and IRF1 protein increased in senescent WI-38 cells (*Figure 2—figure supplement 1F and G*), and silencing IRF1 decreased *TNFSF13B* pre-mRNA levels in WI-38 cells (*Figure 2J*). Together, these findings support a role for the interferon pathway, and the TF IRF1 in particular, in increasing the transcription of *TNFSF13B* mRNA in senescent cells.

## Transcriptomic and proteomic analyses reveal BAFF-dependent roles in senescence

To identify the biological processes regulated by BAFF during senescence in monocytes, we performed transcriptomic and proteomic analyses in senescent THP-1 cells that expressed either normal or reduced levels of BAFF. Given that proliferating THP-1 cells express detectable levels of BAFF, and that BAFF rapidly increases in the initial days after IR treatment (*Figures 1C and 2F*), we silenced BAFF using siRNAs directed at *TNFSF13B* mRNA in proliferating THP-1 cells, and then induced IR treatment the next day (*Figure 3A*). RNA-seq analysis (GSE213993) performed 72 hr after IR revealed 898 transcripts that were highly upregulated by IR but repressed in BAFF-silenced cells, suggesting that induction of these transcripts was dependent on the presence of BAFF (*Figure 3B* and *Figure 3—source data 1*). Searching these transcripts in the Molecular Signatures Database (MSigDB) revealed enrichments in pro-inflammatory gene sets, including those implicated in TNF signaling, NF-κB activation, the IL6/JAK/STAT3 pathway, and the interferon response (*Figure 3C*). Searching the same transcripts in the Gene ontology (GO) 'Biological process' (*Figure 3D*) revealed enrichments in immune biological processes, including leukocyte activation and myeloid activation, while searching for them in the GO 'Cellular component' showed an enrichment in secretory granules and secretory vesicles, suggesting that BAFF could be involved in inflammatory and secretory pathways in senescence. Complementing this notion, transcripts encoding cytokines (IL1B [IL-1β], CCL2, TNFAIP6/TSG-6), chemokines (CXCL1, CXCL2, CXCL8), and alarmins (S100A8, S100A9) were less abundant in BAFF-silenced cells, as assessed in the RNA-seq dataset (*Figure 3E*). Interestingly, the proteins encoded by mRNAs that were reduced in irradiated cells and elevated in BAFF-depleted cells further suggested a role for BAFF in protein localization and cell cycle progression (*Figure 3—figure supplement 1A and B* and *Figure 3—source data 1*).

Proteomic analysis (*Figure 3—source data 2*) revealed several proteins that showed increased abundance in THP-1 cells after IR-induced senescence but were less elevated after BAFF silencing (*Figure 3—figure supplement 1C* and *Figure 3—source data 2*). GO and Reactome pathway analysis of the encoded proteins pointed to the involvement of BAFF in immune activation and leukocyte degranulation, in promoting ROS production and vesicle trafficking (*Figure 3F*, *Figure 3—figure supplement 1C*). The top differentially increased proteins (*Figure 3G and H*) included mediators of inflammation such as NCF1 (Neutrophil Cytosolic Factor 1, also known as p47-phox), Cathepsin B (CATB), and MNDA (Myeloid Nuclear Differentiation Antigen; *Gu et al., 2022*; *Hannaford et al., 2013*; *Holmdahl et al., 2016*). We also found enrichments in proteins involved in intracellular trafficking like RAB1A, RAB1B, and NSF (*Morgan and Burgoyne, 2004*; *Yang et al., 2016*); some of these proteins, like NCF1, ACAA2 (THIM), and CYC1 (CY1), are also implicated in oxidation (*Cao et al., 2008*; *Guo et al., 2017*; *Holmdahl et al., 2016*). The proteins selectively reduced in senescent cells after silencing BAFF suggested a potential role for BAFF in modulating the organization of the nucleosome and cellular checkpoints, possibly through indirect effectors (*Figure 3—figure supplement 1D–F* and *Figure 3—source data 2*). Integrating the data from both the transcriptomic and proteomic analyses from senescent THP-1 cells uncovered robust roles for BAFF in inflammatory pathways and cell activation in senescence.

## BAFF promotes SASP in senescent monocytes

The accumulation of senescent cells in tissues with advancing age has been proposed to have detrimental consequences, as they promote age-related declines and diseases (*Campisi, 2013*; *He and Sharpless, 2017*); accordingly, there is much interest in clearing senescent cells for therapeutic benefit

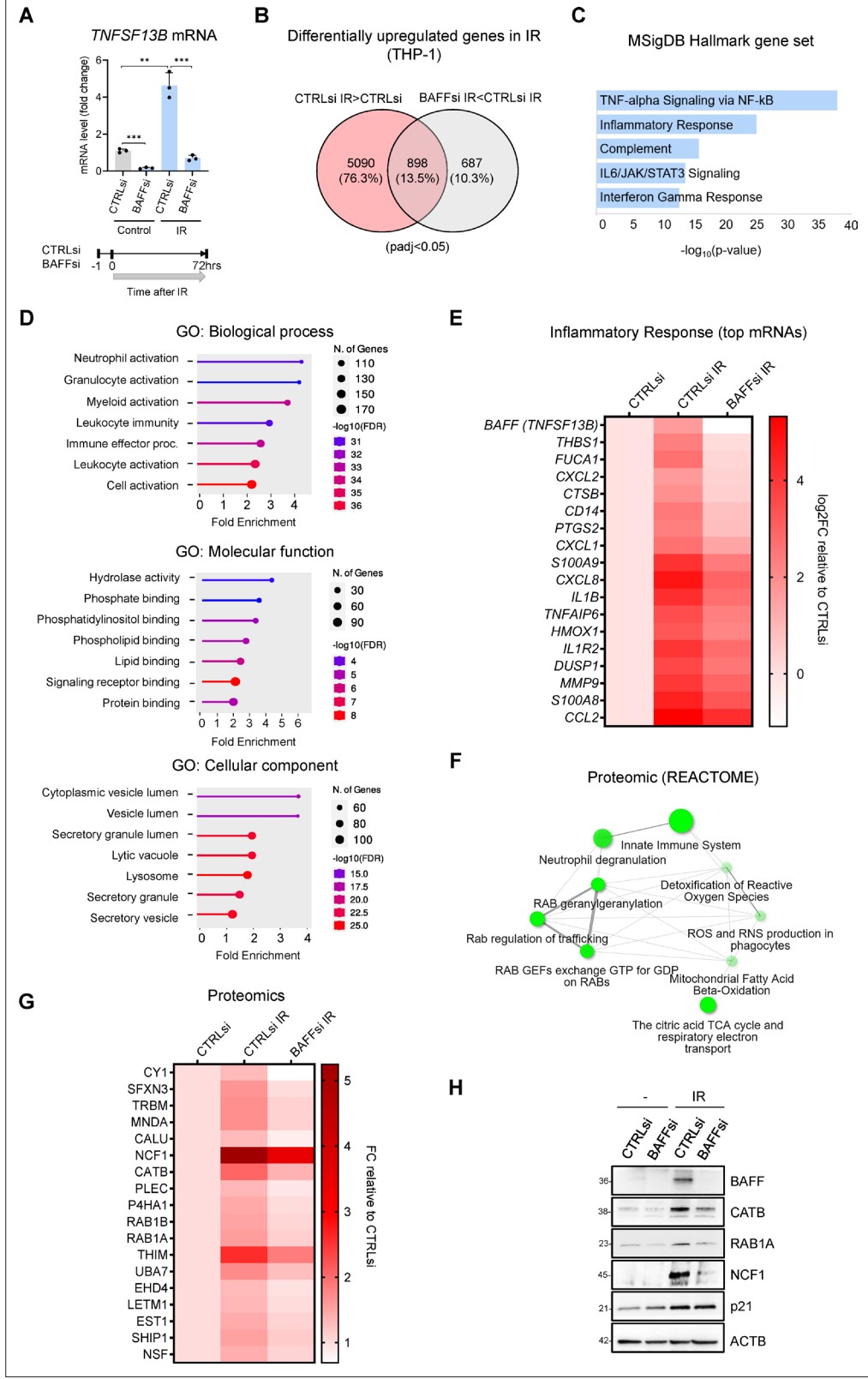

**Figure 3.** Transcriptomic and proteomic analysis in THP-1 cells suggests a role for BAFF in senescence-associated inflammation. (**A**) RT-qPCR analysis of the levels of *TNFSF13B* mRNA in THP-1 cells transfected with CTRLsi or BAFFsi, irradiated 18 hr later (5 Gy), and assessed 72 hr after that. *ACTB* mRNA levels were measured and used for data normalization. *Bottom*, schematic of the timeline of BAFF silencing and exposure to IR in THP-1 cells.

*Figure 3 continued on next page*

*Figure 3 continued*

(**B**) Venn diagram of mRNAs identified by RNA-seq analysis as being differentially upregulated in THP-1 cells transfected with a control siRNA (CTRLsi) or BAFF siRNA (BAFFsi) following exposure to IR (5 Gy, collected 72 hr later). Red circle: mRNAs upregulated in CTRLsi exposed to IR (CTRLsi IR) relative to non-irradiated cells (CTRLsi). Gray circle: mRNAs less induced in BAFFsi cells exposed to IR (BAFFsi IR) relative to CTRLsi exposed to IR (CTRLsi IR). A complete list of genes from the RNA-seq analysis is available (GSE213993 and *Figure 3—source data 1*); padj <0.05. (**C**) Molecular Signatures Database (MSigDB) hallmark gene set summarizing the differentially expressed genes obtained from (**B**); diagram was created with EnrichR and gene sets were ordered by *p* value. (**D**) Gene ontology (Biological Processes, Molecular Function and Cellular Component) of the differentially upregulated mRNAs identified in (**B**). (**E**) Heatmap of the differentially upregulated genes identified in (**B**) and included in the GO terms 'Leukocyte activation', and 'Immune effector process', as well as those present in the MSigDB Hallmark term 'Inflammatory Response'. Values are averages of duplicates. Top genes upregulated in IR were sorted according to their greater reduction after BAFF silencing. Data are shown as Log2FC in gene expression relative to untreated cells (CTRLsi: log2FC = 0). (**F**) Reactome network showing the most highly enriched categories of proteins differentially upregulated in THP-1 cells transfected with control (CTRLsi) or BAFF-directed (BAFFsi) siRNAs and the next day exposed to 5 Gy IR or left untreated and collected 72 hr later (complete proteomic datasets are in *Figure 3—source data 2*; Cutoff fold change was 1.3). Two pathways (nodes) are connected if they share 10% or more proteins. Darker nodes are more significantly enriched protein sets; larger nodes represent larger protein sets. Thicker edges represent more overlapped proteins. (**G**) Heatmap of the top differentially upregulated proteins between CTRLsi and BAFFsi. Top proteins increased after IR were sorted according to their greater reduction after BAFF silencing. Data are shown as fold change between PSM (peptide spectrum matches) relative to the untreated (CTRLsi: FC = 1). Cutoff: ([FC]>1.3, protein with PSM above 15). A complete list of differentially upregulated proteins is available in *Figure 3—source data 2*. (**H**) Western blot analysis of representative proteins identified from the proteomic analysis in (**G**). p21 was included as a control for senescence and ACTB as a loading control. Diagrams in (**D,F**) were created with ShinyGO.

The online version of this article includes the following source data and figure supplement(s) for figure 3:

**Source data 1.** RNA-seq analysis performed in THP-1 cells transfected with CTRLsi or BAFFsi and treated with IR.

**Source data 2.** Proteomic analysis performed in THP-1 cells transfected with CTRLsi or BAFFsi and treated with IR.

**Source data 3.** Uncropped immunoblots for *Figure 3*.

**Figure supplement 1.** Extended data from *Figure 3*.

---

(*Ge et al., 2021*; *Nelson et al., 2012*). This task is hampered by the intrinsic resistance of senescent cells to apoptosis, and thus we investigated whether BAFF is implicated in the viability of senescent cells by silencing BAFF in THP-1 cells and evaluating cell numbers after treatment with IR. As shown in *Figure 3A*, *TNFSF13B* mRNA was successfully silenced by 72 hr after IR (5 Gy), and the levels of cellular BAFF (*Figure 3H*) followed similar trends. Upon BAFF silencing, we did not observe major changes in cell viability, as determined by flow cytometry analysis (*Figure 4—figure supplement 1A*) and trypan blue exclusion analysis (not shown). Similarly, BAFF silencing did not affect cell growth or metabolic activity, as determined by the XTT assay (*Figure 4—figure supplement 1B*). Together, these findings indicate that BAFF does not play a major role in the viability or growth of THP-1 cells, and thus it may not be a valuable target for senolysis of senescent monocytes.

Silencing BAFF in THP-1 cells reduced the percentage of SA-β-Gal-positive THP-1 cells, possibly affecting other senescence traits (*Figure 4A*). Given that the RNA-seq analysis suggested a role for BAFF in inflammation, and given that senescent monocytes display increased secretion of pro-inflammatory molecules (*Elder and Emmerson, 2020*; *Merino et al., 2011*), we tested if BAFF plays an autocrine role in promoting the SASP. Upon BAFF silencing, RT-qPCR analysis revealed that the levels of mRNAs encoding pro-inflammatory factors (e.g. *IL8*, *IL1B*, *MMP9*, and *CCL2* mRNAs, *Figure 3E*, *Figure 4B*) increased with IR but decreased by silencing BAFF. Measurement of the levels of cytokines, chemokines and metalloproteases by multiplex ELISA and cytokine array (*Figure 4C*, *Figure 4—figure supplement 1C*) revealed a general reduction in SASP factors after BAFF silencing in THP-1 cells. The reduced SA-β-Gal activity and SASP factor levels after BAFF silencing in THP-1 cells was confirmed by testing individual BAFF siRNAs (*Figure 4—figure supplement 1D–F*).

To further evaluate if BAFF regulates the SASP, we electroporated THP-1 cells with an empty vector (pCTRL) or BAFF-encoding vector (pBAFF; Materials and methods) to overexpress BAFF and investigate the changes in SASP. The elevated expression of BAFF, as assessed by western blot analysis and ELISA (*Figure 4D and E*), led to increased secretion of multiple SASP factors, especially IL8, MMP9,

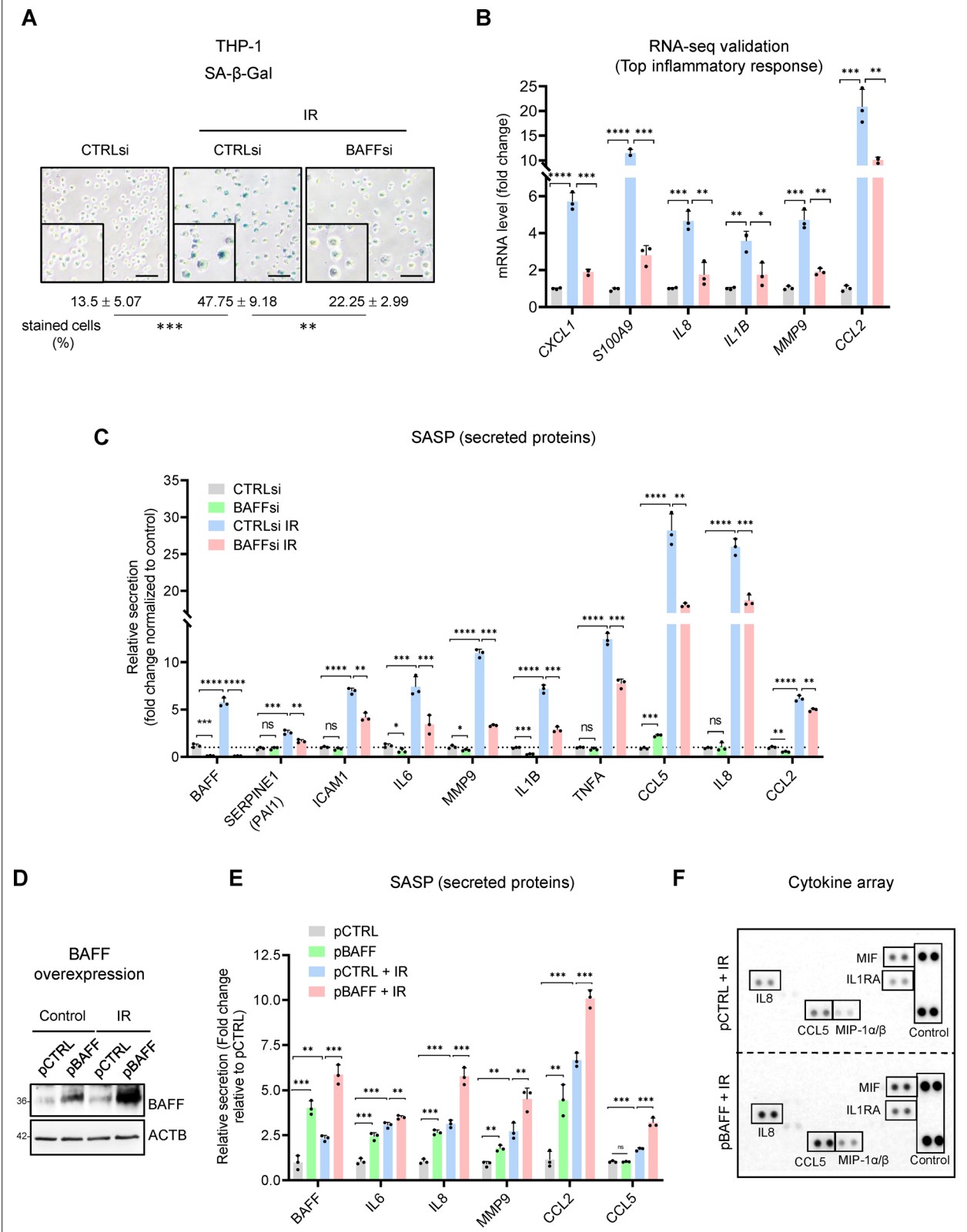

**Figure 4.** BAFF silencing reduces senescence traits in irradiated monocytes, BAFF overexpression increases SASP secretion. (**A**) SA-β-Gal activity assay in THP-1 cells transfected with a control siRNA (CTRLsi) or BAFF siRNA (BAFFsi), following exposure to IR (5 Gy, collected 72 hr later); SA-β-Gal-positive cells (% of total cells) were quantified by percentage (%) of positively stained cells as described (Materials and methods). Scale bars, 100 μm. (**B**) RT-qPCR analysis of a subset of differentially expressed mRNAs identified through RNA-seq analysis (***Figure 3E***) and encoding SASP factors. *ACTB* mRNA

*Figure 4 continued on next page*

*Figure 4 continued*

levels were measured and used for data normalization. (**C**) Relative levels (fold change) of secreted cytokines and chemokines in THP-1 cells processed as in (**A**), as measured by multiplex ELISA 72 hr after IR (5 Gy). (**D,E**) THP-1 cells were transfected for 16 hr with a control plasmid (pCTRL) or a plasmid to express BAFF (pBAFF) and then were either left untreated or treated with IR (5 Gy); 72 hr later, whole-cell lysates were studied by western blot analysis (**D**), and multiplex ELISA assay to detect BAFF and additional SASP factors (**E**). Secretion levels are shown as relative fold change compared to pCTRL group. (**F**) Cytokine array analysis performed on the media of THP-1 cells that were transfected with Control (pCTRL) or BAFF (pBAFF) plasmids, treated with IR (5 Gy) 18 hr later, and assayed 72 hr after that. Reference control spots are present on each individual array. Media were collected from $2\times10^6$ cells per sample. Significance (*, p<0.05; **, p<0.01; ***, p<0.001; ****, p<0.0001) was assessed by Student's *t*-test.

The online version of this article includes the following source data and figure supplement(s) for figure 4:

**Source data 1.** Uncropped western blots for *Figure 4*.

**Figure supplement 1.** Extended data from *Figure 4*.

**Figure supplement 1—source data 1.** Uncropped western blots.

CCL2, and CCL5 (*Figure 4E and F*). Furthermore, THP-1 cells displayed small but significant increases in secreted SASP when treated with soluble recombinant human BAFF (60-mer BAFF, Adipogen; *Figure 4—figure supplement 1G*). These differences were more evident in proliferating cells than in senescent cells, perhaps because ectopic soluble BAFF might only play a limited role in SASP secretion compared to the endogenous or cell membrane-associated BAFF. Overall, BAFF silencing and overexpression experiments suggest that BAFF does not play a role in the viability of senescent monocyte-like THP-1 cells but is important for maintaining the SASP trait.

## BAFF promotes the activation of NF-κB in early senescent THP-1 cells

To study how BAFF promotes inflammation in THP-1 cells, we analyzed the expression of the three known BAFF receptors (BAFFR, TACI, and BCMA) in THP-1 cells. In cells subjected to IR followed by culture for 6 days, western blot analysis indicated that all three receptors increased in senescent cells (*Figure 5A*); additionally, flow cytometry analysis indicated that all three receptors increased on the surface of senescent cells (*Figure 5B*). In sum, the three receptors involved in BAFF signaling via canonical and non-canonical NF-κB pathways (*Afzali et al., 2021*; *Matson et al., 2020*; *Nagel et al., 2014*; *Smulski and Eibel, 2018*) were expressed in senescent THP-1 cells. To find if all receptors were involved in SASP reduction upon BAFF silencing, we silenced them individually (*Figure 5—figure supplement 1A*) in THP-1 cells and triggered IR senescence. Subsequent analysis revealed that silencing BAFFR reduced the expression of key mRNAs encoding SASP factors (*Figure 5C*), while silencing BCMA or TACI did not reduce the expression levels of mRNAs encoding proinflammatory proteins, and instead slightly increased them (*Figure 5C*), suggesting that BAFFR might be the primary BAFF receptor responsible for SASP activation, while BCMA and TACI may possibly function to suppress the SASP. Of note, BAFFR is known to be specific for BAFF, while BCMA and TACI are also known receptors for another cytokine, APRIL, not investigated here (*Afzali et al., 2021*; *Matson et al., 2020*; *Nagel et al., 2014*; *Smulski and Eibel, 2018*); future studies are necessary to define in detail the role of each individual receptor in senescence.

In keeping with the literature on BAFF receptors pathways (*Afzali et al., 2021*; *Matson et al., 2020*; *Nagel et al., 2014*; *Smulski and Eibel, 2018*), EnrichR analysis of differentially expressed transcripts from the RNA-seq (*Figure 3B*) suggested a role for BAFF in the regulation of TFs NF-κB and STAT3 in THP-1 cells (*Figure 5D*, *Figure 5—figure supplement 1B*). We thus analyzed the DNA-binding activity of multiple NF-κB subunits in senescent THP-1 cells upon BAFF silencing (Materials and methods). IR increased the DNA-binding activity of p65, p50, and p52 in control cells, but not the activities of RelB or c-Rel (*Figure 5E*). Interestingly, the activity of p65 and p50, representing the canonical pathway of NF-κB, was strongly reduced after silencing BAFF (*Figure 5E*). To assess the changes in p65/RelA activity by other methods, we prepared cytoplasmic and nuclear fractions from proliferating and senescent THP-1 cells expressing either normal levels of BAFF or reduced BAFF levels by silencing using siRNAs. Interestingly, in proliferating THP-1 cells, silencing BAFF did not change the nuclear abundance of p65, while in senescent THP-1 cells, silencing BAFF significantly reduced the levels of nuclear p65 (*Figure 5F*). Tubulin and PARP1 were used as markers for proper cytoplasmic and nuclear fractionation, respectively, and Ponceau S was used to stain the membrane to assess total loaded proteins (*Figure 5F*). These findings suggest a role for BAFF in the canonical pathway of NF-κB activation in senescent THP-1 cells.

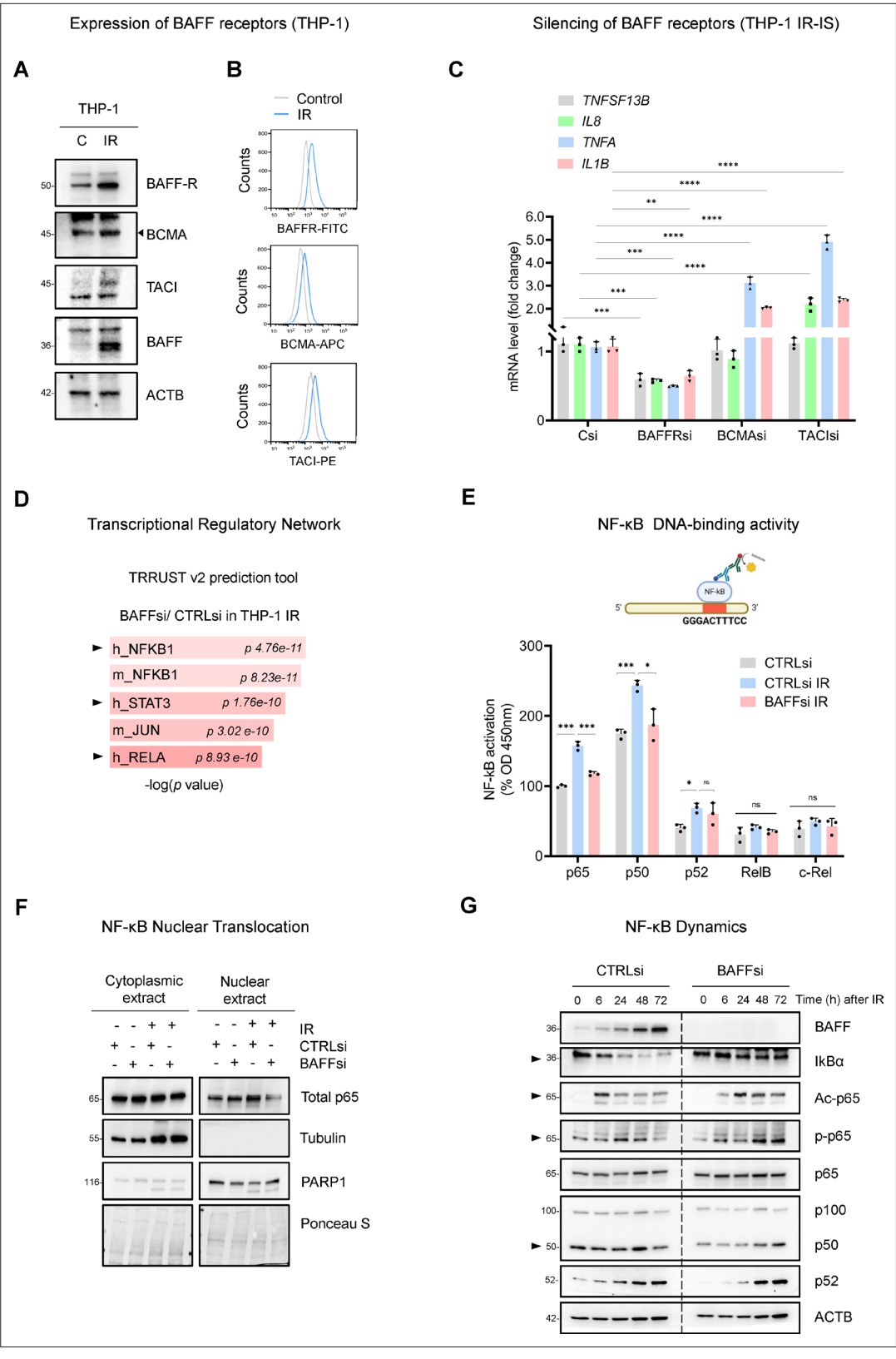

**Figure 5.** BAFF promotes NF-κB activation in IR-treated THP-1 cells. (**A**) Western blot analysis of BAFF receptors in THP-1 cells untreated or irradiated with 5 Gy and cultured for 6 days. Due to the low levels of TACI, contrast was increased on the acquired image. (**B**) Flow cytometry analysis of THP-1 cells expressing surface BAFF receptors 6 days after either no treatment or treatment with IR (5 Gy). (**C**) RT-qPCR analysis of the levels of representative

*Figure 5 continued on next page*

*Figure 5 continued*

SASP mRNAs in THP-1 cells transfected with CTRLsi or siRNA targeting BAFF receptors, irradiated 18 hr later (5 Gy), and assessed 72 hr after that. *ACTB* mRNA levels were measured and used for data normalization. (**D**) EnrichR analysis of the differentially abundant mRNAs obtained from *Figure 3B*. The TRRUST v2 analysis predicts the transcription factors potentially affected by BAFF silencing in THP-1 cells. (**E**) TransAM NF-$\kappa$B Activation Assay (Materials and methods) performed using THP-1 nuclear extracts to evaluate the binding of different NF-$\kappa$B subunits to a DNA consensus sequence (*top*), as measured at 450 nm. Each antibody-specific signal was normalized to the respective blank control. Finally, the basal activity of p65 in the untreated control sample (CTRLsi) was set at 100% and all other values were normalized to it. (**F**) Western blot analysis of p65 levels in nuclear and cytoplasmic fractions of THP-1 cells that were transfected overnight with CTRLsi or BAFFsi and the next day were either left untreated or irradiated and collected 72 hr later. Cytoplasmic and nuclear markers (Tubulin and PARP1, respectively) were included to monitor the fractionation procedure; Ponceau S staining served to assess equal loading and transfer. (**G**) Western blot analysis of the proteins in whole-cell extracts prepared from THP-1 cells that were transfected with either CTRLsi or BAFFsi, then exposed to IR (5 Gy) and assessed at the indicated times. Arrowheads point to signals showing differences in NF-$\kappa$B kinetics after BAFF silencing. Significance (*, $p<0.05$; **, $p<0.01$; ***, $p<0.001$; ****, $p<0.0001$) was assessed by Student's *t*-test.

The online version of this article includes the following source data and figure supplement(s) for figure 5:

**Source data 1.** Uncropped western blots for *Figure 5*.

**Figure supplement 1.** Extended Data from *Figure 5*.

**Figure supplement 1—source data 1.** Uncropped western blots and arrays.

A crucial step in the canonical pathway of NF-κB activation is the degradation of the inhibitory IκB proteins, which precedes NF-κB activation and nuclear translocation (*Hayden and Ghosh, 2008*; *Israël, 2010*). Thus, to further study the role of BAFF on the activation of the canonical NF-κB pathway, we analyzed the pattern of IκBα degradation. As shown in *Figure 5G*, IκBα levels declined rapidly after treatment with IR in control cells, but not in cells in which BAFF was silenced, further supporting a role for BAFF in modulating NF-κB activity.

BAFF was previously shown to activate the canonical NF-κB pathway in monocytes after LPS treatment by affecting p65 acetylation, which is required for DNA binding and full transcriptional activity (*Gardam and Brink, 2014*; *Lim et al., 2017*). Western blot analysis of p65 acetylation relative to total p65 in irradiated THP-1 cells showed a transient peak in p65 acetylation at ~6 hr after irradiation in control cells, and a delayed peak at ~24–48 hr followed by persistently elevated p65 acetylation in BAFF-silenced cells (*Figure 5G*). Similarly, p65 phosphorylation also showed a delayed peak in BAFFsi cells, reaching a maximum at ~24 hr after irradiation in control cells, and at ~48–72 hr in BAFFsi cells. In sum, the influence of BAFF on IκBα degradation, p65 nuclear accumulation and p65 modifications may affect the DNA-binding activity and transcriptional program of NF-κB. Other NF-κB subunits (p50 and p52) changed only modestly as a function of BAFF abundance (*Figure 5G*). Although they may also influence NF-κB activity overall, our results point to a major role of BAFF in the regulation of p65 function.

Finally, to study the impact of BAFF on NF-κB activation in THP-1 cells, we overexpressed BAFF by co-transfecting pBAFF (or the control empty vector pCTRL) with one of three plasmids: a control plasmid driving negligible expression of GFP from a minimal promoter (pMin-GFP), a plasmid driving constitutive expression of GFP from the strong, constitutively active promoter CMV (pCMV-GFP), and a plasmid driving GFP expression from an NF-κB-inducible reporter (pNF-κB-GFP; Cignal Reporter Assay, Qiagen; *Figure 5—figure supplement 1C*). When compared with pCTRL, ectopic overexpression of BAFF after transfecting pBAFF increased GFP production from pNF-κB-GFP, but did not increase GFP production from pMin-GFP or pCMV-GFP (*Figure 5—figure supplement 1C*).

To assess more broadly other pathways, kinases, and TFs that might be regulated by BAFF in senescent monocytes, we examined the results of phospho-protein array analysis. Interestingly, BAFF silencing in THP-1 cells reduced STAT3 phosphorylation (*Figure 5—figure supplement 1D*), in agreement with the EnrichR prediction (*Figure 5D* and *Figure 5—figure supplement 1B*) and with the role of STAT3 in SASP production (*Kojima et al., 2013*). We also observed a reduction in the active forms of ERK1/2 (*Figure 5—figure supplement 1D*), previously implicated in the induction of inflammatory responses in monocytes in response to tissue damage and activation of Toll-like receptor pathways (*Lucas et al., 2022*). Taken together, our findings indicate that BAFF contributes to the activation

of proinflammatory pathways in senescent monocytes, at least in part by activating p65/NF-κB and possibly other signaling pathways including STAT3.

## Transcriptomic analysis in fibroblasts reveals alternative roles for BAFF in senescence

To gain a more complete understanding of the role of BAFF in senescence, we investigated its function in primary fibroblasts, which are well-established models for senescence and, unlike THP-1 cells, express the senescence-relevant protein p53. Given that the expression of *TNFSF13B* mRNA and BAFF protein increased more slowly in fibroblasts relative to THP-1 cells (see *Figure 2F*, *Figure 2— figure supplement 1F*, *Figure 5G*, and *Figure 6—figure supplement 1A and B*), we sequentially transfected siRNAs at days 0 (before IR) and 7, and harvested cells at day 10 to achieve a prolonged depletion of BAFF in WI-38 cells (*Figure 6—figure supplement 1C*).

Given that BAFF silencing reduced senescence and the SASP in THP-1 cells (*Figure 4B and C*), we tested if fibroblasts displayed similar or different characteristics upon BAFF depletion. As shown, silencing BAFF in senescent WI-38 fibroblasts led to a strong decrease in SA-β-Gal staining (*Figure 6A and B*), but the effects on the secretion of SASP factors were mainly restricted to changes in IL6 (*Figure 6C*). We confirmed these results by testing two individual BAFF siRNAs (*Figure 6—figure supplement 1D and E*). Similarly, BAFF silencing in senescent IMR-90 fibroblasts treated with etoposide (50 μM ETO for 72 hr, followed by 7 days in culture, for a total of 10 days) decreased the SA-β-Gal staining and IL6 expression levels (*Figure 6—figure supplement 1F and G*), with no effects on the levels of other SASP factors (not shown).

To examine in greater detail the role of BAFF in senescent fibroblasts, we analyzed gene expression profiles after silencing BAFF in senescent WI-38 cells. The mRNAs that increased or decreased in senescent WI-38 cells as a function of BAFF levels, as identified by RNA-seq analysis [*Figure 6D* and *Figure 6—source data 1* (GSE213993)], highlighted a major role for BAFF in the induction of the p53 pathway (*Figure 6E and F left*) and the repression of transcripts involved in cell cycle progression (*Figure 6E and F right*). RT-qPCR analysis of some of the p53 target genes found in the RNA-seq analysis (*FUCA1*, *CLCA2*, *LIF*, and *CYFIP2* mRNAs; *Figure 6G*, **top**) confirmed that their expression increased in senescent cells and was significantly diminished by BAFF silencing. Similarly, RT-qPCR analysis confirmed that the expression levels of many mRNAs (e.g. *AURKB*, *CDK1*, *MKI67*, *LMNB1*, and *LBR* mRNAs) that were significantly reduced in senescent cells were moderately restored by BAFF silencing (*Figure 6F*, **right**, and *Figure 6G*, **bottom**). These data helped to explain the impairment of fibroblast senescence after BAFF silencing and revealed a role for BAFF in promoting p53 function in primary fibroblasts.

## BAFF promotes p53 expression and activity in p53-proficient senescent cells

The p53 transcription factor is a key regulator of cellular stress responses and senescence by governing gene expression programs and suppressing cell proliferation (*Mijit et al., 2020*). Given that the WI-38 transcriptome suggested that BAFF activated p53 in senescent fibroblasts (*Figure 6E–G*), we analyzed the levels of total and phosphorylated p53 in senescent fibroblasts.

Phosphoarray analysis revealed that BAFF silencing reduced the levels of phosphorylated p53 in WI-38 cells (*Figure 7A*), indicating a possible reduction in transcriptionally active p53, as well as changes in the phosphorylation of ERK1/2, similarly to what we observed in THP-1 cells (*Figure 7A* and *Figure 5—figure supplement 1D*). Paradoxically, while the activation of ERK1/2 promoted cell proliferation in dividing cells, it also promoted growth arrest in senescent cells (*Anerillas et al., 2020*), although further studies are needed to determine the precise role of ERK1/2 in the pathway activated by BAFF. To confirm the role of BAFF in p53 regulation, we measured the levels of total p53 in senescent WI-38 and IMR-90 fibroblasts after BAFF silencing, and we observed a strong reduction of p53 protein (*Figure 7B* and *Figure 7—figure supplement 1A*). Since a role for BAFF in the regulation of p53 function was not previously described, we investigated whether BAFF might signal through its receptors to modulate p53 levels. We detected for the first time the presence of BAFF receptors in WI-38 cells (*Figure 7—figure supplement 1B*), although they were mainly localized intracellularly, rather than on the cell surface, as shown by western blot analysis (*Figure 7—figure supplement 1B*) and confirmed by flow cytometry (not shown); a similar intracellular presence of BAFF receptors was

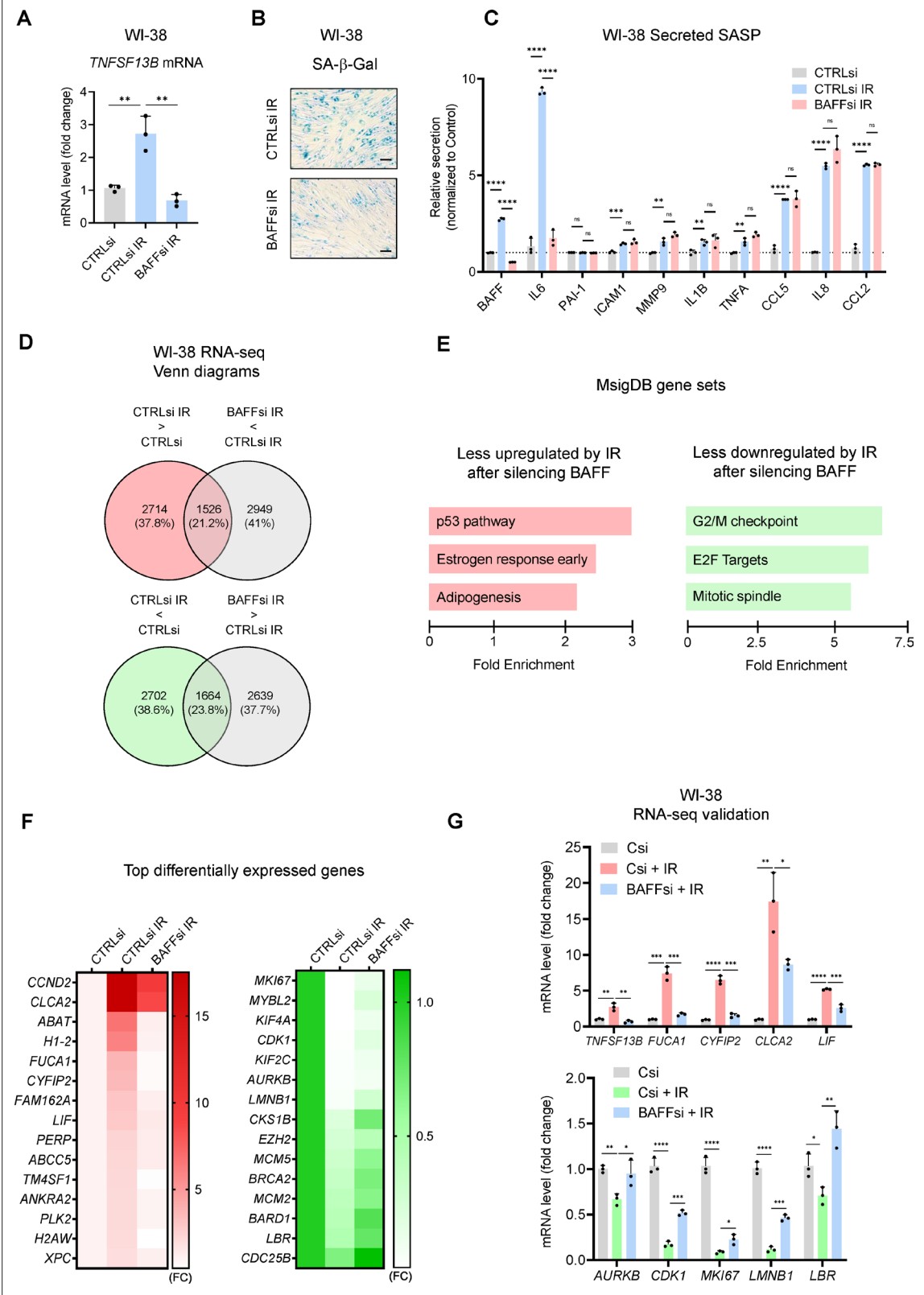

**Figure 6.** Transcriptomic analysis suggests a role for BAFF in p53 activation in senescent WI-38 fibroblasts. (**A**) RT-qPCR analysis of *TNFSF13B* mRNA levels in WI-38 fibroblasts transfected with CTRLsi or BAFFsi, cultured for 18 hr, then left untreated or treated with a single dose of IR (10 Gy), and assayed 10 days later (Materials and methods). *ACTB* mRNA levels were measured and used for data normalization. (**B**) SA-β-Gal activity assay in WI-38 cells treated as in (**A**). Scale bar, 100 μm. (**C**) Levels of BAFF, IL6, and other SASP factors in the culture media of WI-38 cells treated as in (**A**), measured by

*Figure 6 continued on next page*

*Figure 6 continued*

multiplex ELISA. (**D**) WI-38 cells processed as described in (**A**) were subjected to RNA-seq analysis (GSE213993; *Figure 6—source data 1*). *Top*, mRNAs showing increased abundance after IR (red) and mRNAs showing reduced abundance after silencing BAFF (gray) are identified at the intersection (padj <0.05). *Bottom*, mRNAs showing reduced abundance after IR (green) and higher levels after silencing BAFF (gray) are identified at the intersection (padj <0.05). (**E**) MSigDB hallmark gene set enrichment performed on the differentially expressed mRNAs in WI-38 fibroblasts. *Left*, pathways less upregulated by IR after silencing BAFFsi. *Right*, pathways less reduced by IR after silencing BAFFsi. Bars are ordered according to the fold enrichment of individual pathways. (**F**) Heatmaps of the top differentially expressed mRNAs from the top gene sets in (**E**). Data are average of two values and are shown as fold change (FC) relative to the control (CTRLsi FC = 1). *Left*, top mRNAs selectively induced by IR and reduced after silencing BAFF. *Right*, top mRNAs selectively reduced by IR but remaining expressed after silencing BAFF. (**G**) Validation by RT-qPCR analysis of representative mRNAs identified by RNA-seq analysis in (**D**) and listed in the heatmaps in (**F**). *ACTB* mRNA levels were used for data normalization. Significance (*, p<0.05; **, p<0.01; ***, p<0.001; ****, p<0.0001) was assessed by Student's *t*-test.

The online version of this article includes the following source data and figure supplement(s) for figure 6:

**Source data 1.** RNA-seq analysis performed in WI-38 fibroblasts transfected with CTRLsi or BAFFsi and treated with IR.

**Figure supplement 1.** Extended data from *Figure 6*.

**Figure supplement 1—source data 1.** Uncropped western blots.

reported in other cell types and conditions (*Dimitrakopoulos et al., 2019*; *Fu et al., 2009*; *Garcia-Carmona et al., 2018*; *Hatzoglou et al., 2000*). Therefore, it is possible that BAFF may signal through a non-canonical pathway in fibroblasts. Next, we individually silenced the three BAFF receptors in senescent WI-38 cells. Interestingly, silencing of BAFFR yielded a strong reduction in total p53 levels (*Figure 7—figure supplement 1C*), while silencing of BCMA or TACI did not reduce p53 levels, and instead increased total p53 levels, similarly to what we observed for TACI and BCMA in THP-1 cells when measuring SASP factor levels (*Figure 5C*). Therefore, it is possible that these two receptors may antagonize the primary BAFF signaling, or they may be activated by other ligands like APRIL, not investigated here (*Afzali et al., 2021*; *Matson et al., 2020*; *Nagel et al., 2014*; *Smulski and Eibel, 2018*).

Next, we investigated whether BAFF promotes p53 levels and activity not only in senescent fibroblasts, but also in senescent monocytes. Unfortunately, THP-1 cells do not express p53 due to a 26-nucleotide deletion in the p53 coding sequence that prevents p53 production (*Sugimoto et al., 1992*; *Figure 7C*); therefore, we sought to test primary monocytes and macrophages, but the inherent difficulty of transfecting them made this effort impossible. Instead, we studied murine RAW 264.7 cells, widely used as a model of p53-proficient monocyte/macrophage-like cells (*Hassan et al., 2005*). BAFF silencing in RAW 264.7 cells was followed by treatment with 5 Gy IR (Materials and methods), and 72 hr later we assessed the levels of total and phosphorylated p53. Importantly, BAFF silencing reduced p53 phosphorylation levels (at S15) in RAW 264.7 cells, although total p53 levels were unchanged (*Figure 7C* and *Figure 7—figure supplement 1D*). Furthermore, BAFF silencing reduced the levels of secreted general SASP factors in murine RAW 264.7 cells, similarly to what we observed in the human monocytic-like THP-1 cells (*Figure 7—figure supplement 1E* and *Figure 4C*).

Overall, our data have uncovered some shared roles for BAFF in senescent monocytes and senescent fibroblasts, including the requirement for BAFF receptors, the loss of both SA-β-Gal activity and IL6 after silencing BAFF, and the reduction in the activation of ERK1/2. However, our data also highlight interesting differences in the impact of BAFF on the senescence of monocytes and fibroblasts (*Figure 7D*). In monocyte- and macrophage-like cells (THP-1 and RAW 264.7 cells), BAFF widely promoted SASP production (*Figure 7—figure supplement 1E* and *Figure 4C*), while we only observed minor changes in SASP after silencing BAFF in senescent fibroblasts (*Figure 6C* and *Figure 6—figure supplement 1D, F*). In contrast, in senescent fibroblasts and in RAW 264.7 cells, silencing BAFF reduced p53 levels and/or phosphorylation (*Figure 6E*, *Figure 7A–C*). Together, our results indicate that the senescence-associated protein BAFF is jointly elevated across senescence paradigms but its functional impact on senescence programs varies depending on the cellular context (*Figure 7D*).

## Discussion

Over the lifetime, internal and external factors, like replicative exhaustion, oxidants, viral infection, inflammation, and cancer therapies can cause sublethal cell damage that leads to cellular senescence. Despite their persistent growth arrest, senescent cells remain metabolically active and release a variety

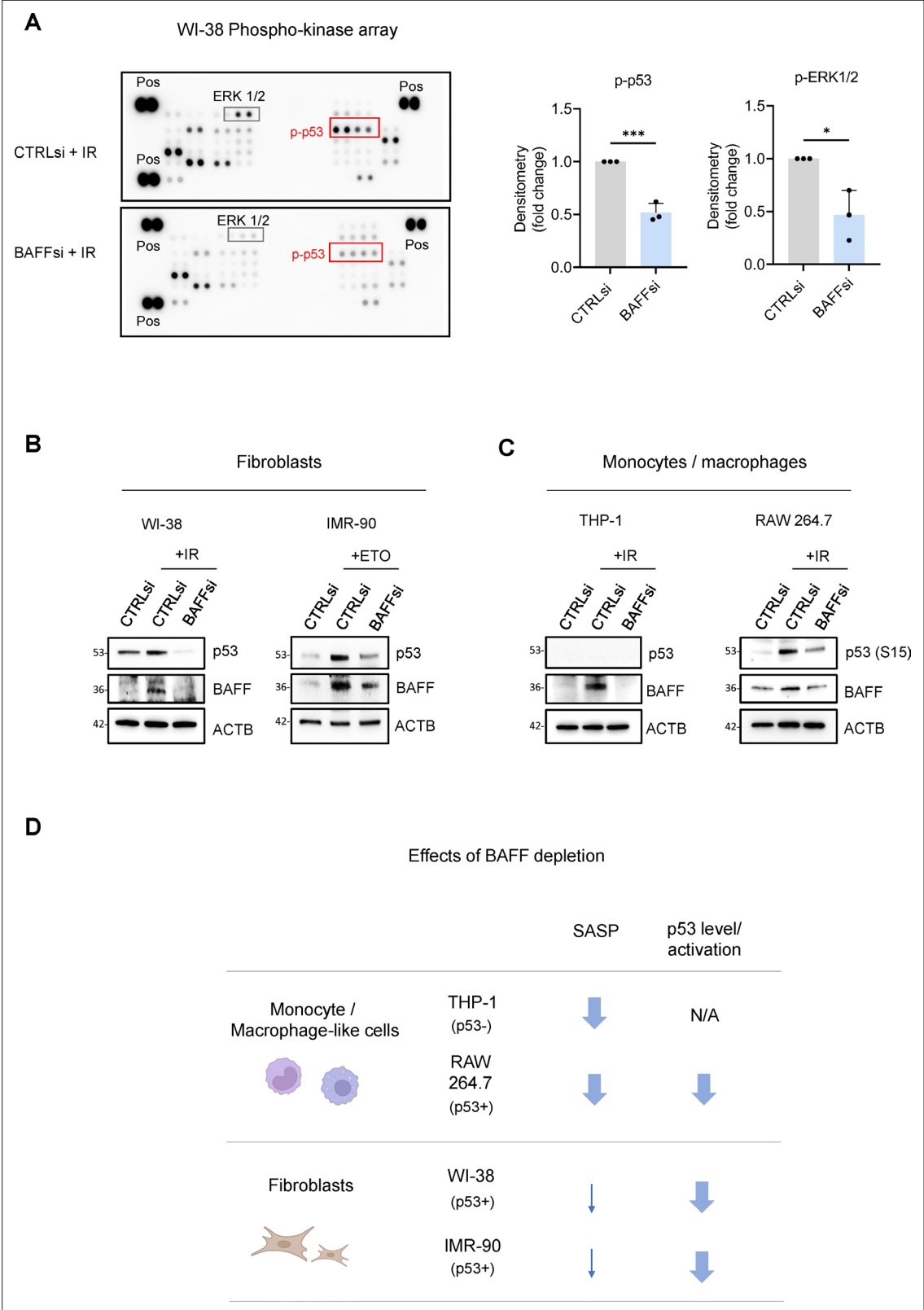

**Figure 7.** BAFF silencing reduces senescence traits and p53-dependent genes in fibroblasts. (**A**) *Left,* phosphoarray analysis of whole-cell lysates prepared from WI-38 fibroblasts transfected with CTRLsi or BAFFsi, cultured for 18 hr, and then either left untreated or treated with a single dose of IR (10 Gy), and harvested 10 days later. The phosphoarray included positive control ('Pos') reference signals. *Right*: densitometry analysis of differentially phosphorylated and statistically significant proteins, averaged from three independent replicates (see *Figure 7—source data 1*). Significance in

Figure 7 continued

different panels (*, p<0.05; ***, p<0.001) was assessed by Student's *t*-test. (**B**) Whole-cell lysates were prepared from WI-38 and IMR-90 fibroblasts transfected with CTRLsi or BAFFsi, treated with a single dose of IR (10 Gy, WI-38) or ETO (50 µM, IMR-90), and collected 10 days later. The levels of p53, BAFF, ACTB were assessed by western blot analysis. (**C**) Whole-cell lysates were prepared from THP-1 and RAW 264.7 cells transfected with CTRLsi or BAFFsi, treated with a single dose of IR (5 Gy) and collected 72 hr later. The levels of p53 (total or phosphorylated), BAFF, and ACTB were assessed by western blot analysis. (**D**) Schematic highlighting the different effects of BAFF depletion on the senescent phenotype of primary fibroblasts (WI-38, IMR-90) and monocyte/macrophage-like cells (THP-1, RAW 264.7). Arrows indicate a reduction in the observed phenotype. N/A: not applicable.

The online version of this article includes the following source data and figure supplement(s) for figure 7:

**Source data 1.** Uncropped blots and arrays for *Figure 7*.

**Figure supplement 1.** Extended data from *Figure 7*.

**Figure supplement 1—source data 1.** Uncropped western blots.

of cytokines, chemokines, growth factors, and metalloproteinases—a trait collectively known as the SASP. The negative consequences of accumulating senescent cells in tissues are mainly linked to the production of these molecules that promote a pro-inflammatory microenvironment, which in turn activates an immune response and triggers tissue remodeling with loss of normal tissue architecture. Accordingly, senescent cells are believed to participate in the persistent inflammation that develops with age ('inflammaging') and is associated with age-related pathologies like cancer, cardiovascular and neurodegenerative diseases, and dysfunctions of lung, liver, and pancreas (*Franceschi et al., 2007*; *McHugh and Gil, 2018*; *Muñoz-Espín and Serrano, 2014*).

Despite a heightened interest in cell senescence, the development of translational approaches to identify and clear senescent cells has been hindered by an incomplete understanding of the molecular markers of senescent cells, including those that are universally present in all senescent cells and those that define specific senescent cell subgroups. Here, we focused on BAFF, a cytokine previously predicted to be increased across models of cell senescence in culture (*Casella et al., 2019*). We found it elevated not only in cultured cell models but also in a model of senescence in mice (*Figure 1*). We propose that BAFF is a novel potential biomarker of senescence both in culture and in vivo, as it is easily measurable using sensitive methods like RT-qPCR analysis and ELISA.

BAFF has been extensively studied in immunology, as it plays a primary role in B-cell maturation and survival (*Mackay and Browning, 2002*; *Schiemann et al., 2001*). Besides its role in the homeostasis of the immune system, BAFF was also implicated in autoimmune diseases like lupus and multiple sclerosis, in part associated with its pro-inflammatory function (*Davidson, 2010*; *Moisini and Davidson, 2009*). We validated the increase of *TNFSF13B* mRNA and protein in multiple senescent models (*Figure 1* and *Figure 1—figure supplement 1*) and found that IRF1 transcriptionally elevated *TNFSF13B* mRNA levels in senescence (*Figure 2* and *Figure 2—figure supplement 1*). Given that BAFF is mainly produced by monocytes (*Yoshimoto et al., 2020*; *Yoshimoto et al., 2011*), we focused on the induction of BAFF in the monocytic cell line THP-1. Transcriptomic and proteomic analyses suggested a role for BAFF in monocyte activation and inflammation in senescence (*Figure 3* and *Figure 3—figure supplement 1*), as BAFF depletion led to a striking reduction in the production of SASP factors in irradiated THP-1 cells, while BAFF overexpression or ectopic addition of BAFF increased the secretion of pro-inflammatory molecules (*Figure 4* and *Figure 4—figure supplement 1*). THP-1 cells expressed all three BAFF receptors (BCMA, TACI, and BAFFR), and downstream signaling pathways culminated with the activation of the NF-κB component p65/RelA after DNA damage (*Figure 5* and *Figure 5—figure supplement 1*). The activation of this pathway could explain, at least in part, the reduction of SASP factor production after silencing BAFF in THP-1 cells. Interestingly, however, in senescent fibroblasts, BAFF did not have a strong impact on SASP factor biosynthesis, except for IL6 production; instead, BAFF appeared to modulate the levels and activity of the senescence-associated TF p53 (which is not expressed in THP-1 cells) (*Figure 6* and *Figure 6—figure supplement 1*). Further analysis revealed that BAFF promotes p53 activation not only in fibroblasts, but also in p53-proficient monocyte/macrophage-like RAW 264.7 cells (*Figure 7* and *Figure 7—figure supplement 1*). Finally, BAFF influenced key senescence traits in both monocytes and fibroblasts, including SA-β-Gal activity, IL6 production, and ERK1/2 modulation (*Figure 4A–C*, *Figure 6A–C*, *Figure 6—figure supplement 1D–G*, *Figure 7A*, *Figure 5—figure supplement 1D*). A schematic summarizing these findings and a proposed model for BAFF regulation and role in senescence is offered (*Figure 8*).

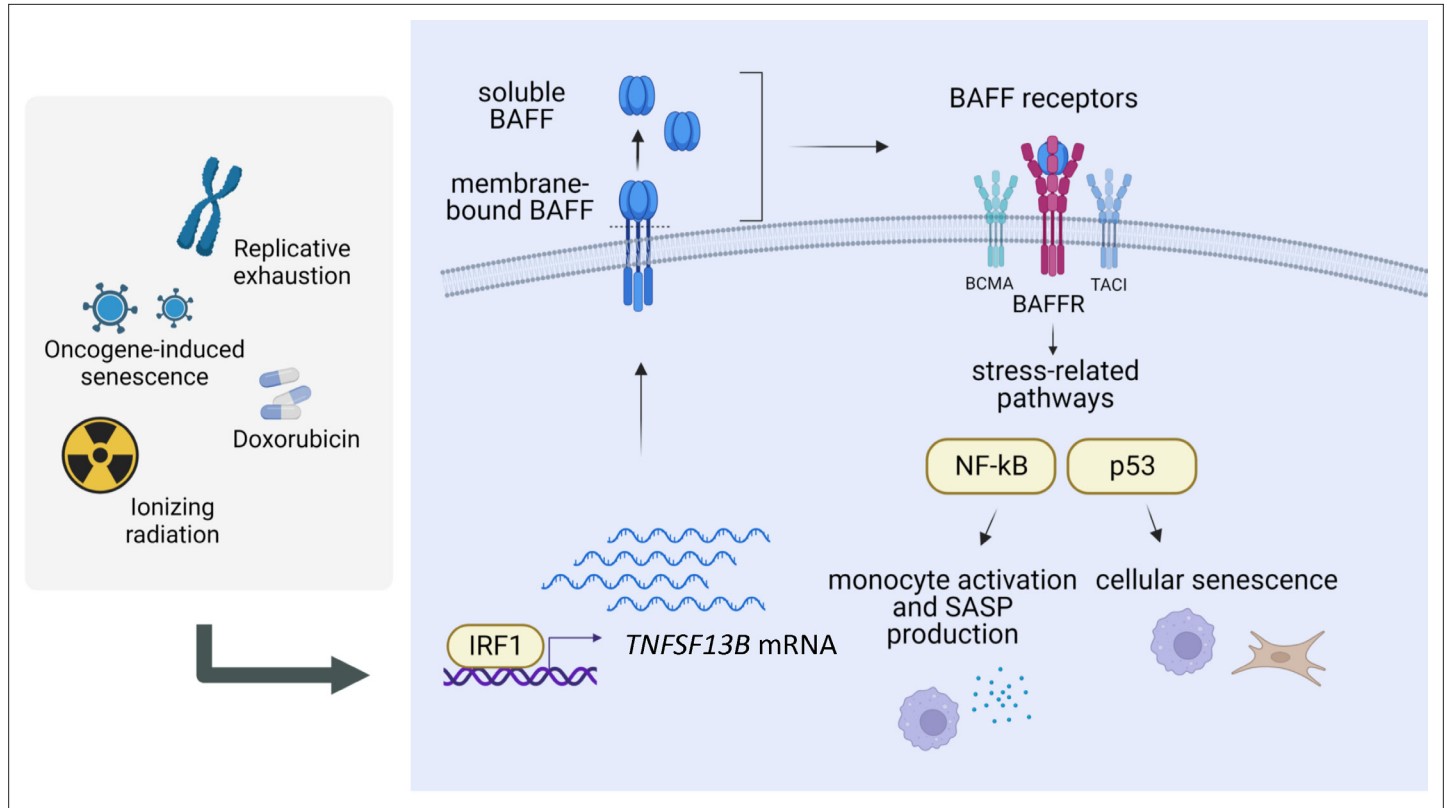

**Figure 8.** Regulation and role of BAFF in senescence. Proposed model for the regulation and role of BAFF in senescence (created using BioRender). Following DNA damage, the TF interferon-regulated factor IRF1 induces the transcription of *TNFSF13B* mRNA. The protein BAFF is translated and inserted into the plasma membrane, where it can be further processed into a secreted form. Both forms of BAFF are increased in senescence, and both have been previously reported to be functional and capable of activating BAFF receptors (BAFFR, TACI, BCMA), which in turn stimulate stress-related pathways in a cell type-dependent manner, with a predominant activation of the NF-$\kappa$B pathway in monocytic-like cells, and the p53 pathway in primary fibroblasts. Therefore, BAFF may have pleiotropic actions on senescence-associated phenotypes in different cell types. We propose that BAFF is a novel biomarker of senescence and a regulator of different senescence traits.

These results agree with the emerging view that senescence is a heterogeneous response across different tissues and cell types, varies according to the inducers of senescence and the time elapsed since senescence was triggered, and is robustly influenced by the microenvironment (*Cohn et al., 2022*). The discovery of factors shared across senescence paradigms and factors specific for select senescent programs can allow more precise interventions when targeting therapeutically this complex cell population.

To guide future strategies targeting BAFF, it will be important to identify the precise signaling mediators that connect activated BAFF receptors to p53 function in primary senescent cells. It will also be important to study the specific contribution of membrane-bound BAFF relative to secreted BAFF in the implementation of the SASP and other senescence traits. Our preliminary data in THP-1 cells using recombinant BAFF suggest that membrane-bound BAFF has a predominant role over soluble BAFF, as we observed only a minor effect of the recombinant BAFF on the induction of SASP (*Figure 4—figure supplement 1G*), although further studies are necessary to confirm this hypothesis. Unexpectedly, however, BAFF-neutralizing agents strongly induced a pro-inflammatory response in THP-1 cells (not shown), supporting a possible reverse signaling triggered by molecules that interact with the membrane-bound BAFF ('out-to-in' signaling). However, reverse signaling for BAFF remains a point of debate (*Jeon et al., 2010*; *Nys et al., 2013*; *Zhang et al., 2015*) and further studies are necessary to confirm these possibilities in senescent cells.

The strategies adopted to eliminate senescent cells or reduce their negative effects include use of senotherapeutic compounds, some of which have entered clinical trials. Senolytic drugs preferentially induce the death of senescent cells over non-senescent cells, while senomorphic compounds modulate the senescent phenotype and SASP production without eliminating senescent cells (*Niedernhofer and Robbins, 2018*; *Zhang et al., 2015*). Given that BAFF is a SASP factor and plays a key role in modulating

senescence-associated phenotypes like the SASP, BAFF could be exploited as a potential target of senomorphic therapy or perhaps even as a marker of the efficacy of senomorphic interventions, not only in laboratory settings in culture, but also in animal models and possibly in human trials. The heterogeneous responses observed between monocytes in fibroblasts underscore the importance of studying the role of BAFF in paradigms of senescence involving other cell types (epithelial cells, myoblasts, adipocytes, endothelial cells, hepatocytes, glial cells, etc.) and different senescence inducers. Specifically, since B lymphocytes have been established as a main target cell for BAFF (*Mackay and Browning, 2002*; *Schiemann et al., 2001*), it will be critical to study the role of BAFF in senescent B cells and age-related immunosenescence. Future work is warranted to test comprehensively if modulators of BAFF activity have therapeutic value in disease states in which senescent cells are detrimental.

## Materials and methods

### Cell culture and senescence induction

Human acute monocytic leukemia THP-1 (ATCC, TIB-202) cells were cultured in RPMI-1640 medium (Gibco) supplemented with heat-inactivated 10% fetal bovine serum (FBS, Gibco), and 1% penicillin and streptomycin (Gibco). Human primary WI-38 and IMR-90 diploid fibroblasts (Coriell Institute, AG06814-J, I90-79) were cultured in Dulbecco's modified Eagle's medium (DMEM, Gibco) supplemented with 10% FBS, 1% antibiotics, and 1% non-essential amino acids (Gibco). The karyotype of WI-38 and IMR-90 fibroblasts is 46,XX (normal diploid cells, female). Human coronary artery vascular smooth muscle cells (hVSMCs) were obtained from LifeLine Cell Technology and were cultured in VascuLife SMC Medium Complete Kit (LifeLine Cell Technology, FC-005). RAW 264.7 cells (ATCC, TIB-71; karyotype was not specified) were cultured in DMEM supplemented with 10% FBS and 1% antibiotics. RAW 264.7 cells were established from an ascites of a tumor induced in a male mouse by intraperitoneal injection of Abelson Leukaemia Virus (A-MuLV). All cultures were maintained in an incubator at 37 °C and 5% $CO_2$ and tested negative for mycoplasma. Senescence was induced by exposure to different doses of ionizing radiation (IR) (5 Gy for THP-1 cells and RAW 264.7 cells; 10 Gy for WI-38 and hVSMCs) followed by incubation for the times indicated in text and figure legends. Etoposide (ETO)-induced senescence was achieved by treating IMR-90 cells with 50 µM ETO for 72 hr, followed by culture in fresh media for 7 days (10 days since adding ETO). Replicative senescence of WI-38 fibroblasts was achieved by passaging proliferating cells [typically at population doubling level (PDL) of 20–24] until they stopped proliferating (typically PDL >50). Doxorubicin-induced senescence was triggered by treating cells with a single dose of doxorubicin (DOX; 250 nM for WI-38 cells, 10 nM for THP-1 cells) and culturing for the numbers of days specified in each case. Oncogene-induced senescence (OIS) was induced by transducing the cells for 18 hr with a lentiviral vector that expressed RasG12V; lentiviral vectors (control and RasG12V) were reported (*Basu et al., 2018*). The inhibitors of the interferon pathway Ruxolitinib (Ruxo, used at 1 µM), MRT67307 (MRT67, used at 5 µM), and BX-795 (used at 5 µM) were from InvivoGen. XTT assay and BrdU proliferation assays cells were performed in 96-well plates with specific kits, following the manufacturers' instructions (XTT Assay Kit, Abcam, ab232856; BrdU Cell Proliferation Assay Kit, CST, #6813).

### Transfection and nucleofection

Silencing interventions in all the cells analyzed were performed by transfecting 50 nM of siRNAs using Lipofectamine (RNAiMax, Invitrogen) following the manufacturer's instructions. Non-targeting siRNA pool was purchased from Horizon Discovery (cat. D-001206-14-20). For BAFF silencing in human cells, we first validated two individual siRNAs, BAFFsi #1 (BAFFsi D-017586-01-0005) and BAFFsi #2 (BAFFsi D-017586-03-0005); where indicated, the two siRNAs were used as a pool (referred to as BAFFsi). For BAFF silencing in murine RAW 264.7 cells, we used a SMARTpool consisting of 4 siRNAs (Horizon Discovery, cat. M-046829-00-0010); siRNAs (cat. D-001206-14-20) served as a non-targeting control. For IRF1 silencing, we used two different pools of control and IRF1 siRNAs. Pools #1 (*Figure 2*) were from Santa Cruz Biotechnology (IRF1 siRNA, cat. sc-35706, and control siRNA, cat. sc-37007) and pools #2 (*Figure 2—figure supplement 1*) were from Horizon Discovery (IRF1 siRNA, cat. M-011704-01-0005, and control siRNA, cat. D-001206-14-20). To silence BAFF receptors, we used Horizon Discovery SMARTpools (M-008095-00-0005, M-013424-00-0005, M-011217-02-0005).

THP-1 and RAW 264.7 cells were transfected at a density of $3×10^5$ cells/ml; 18 hr later they were treated with IR (5 Gy) in PBS, given fresh medium, and returned to the incubator. WI-38 cells were

transfected at a density of $2×10^5$ cells/well of a six-well plate; 18 hr later, they were treated with 10 Gy IR in PBS, given fresh medium, and returned to the incubator. IMR-90 cells were transfected at a density of $2×10^5$ cells/well; 18 hr later, they were treated with 50 µM etoposide (ETO) for an additional 72 hr, followed by culture for 7 days (10 days total since adding ETO). The plasmid vector for BAFF (pBAFF) was from GenScript (OHu22261) and pcDNA3.1(+) plasmid was used as empty vector (pCTRL). BAFF overexpression in THP-1 cells was achieved by nucleofection (Nucleofector kit V, Lonza, program V-001) following the manufacturer's instructions. For each reaction, we added 0.5 µg of plasmid to $10^6$ THP-1 cells; 18 hr later, cells were treated with 5 Gy IR in PBS, given fresh media, returned to the incubator, and analyzed at the times specified in the figure legends. To co-transfect overexpression plasmids and NF-κB reporters, we used lipofectamine LTX with PLUS reagent (A12621), following the manufacturer's instructions. GFP reporter plasmids for NF-κB (pNF-κB-GFP) and specific controls (minimal promoter plasmid pMin-GFP and constitutively active promoter plasmid pCMV-GFP) were from Qiagen (Cignal Reporter Assay Kit, ID: 336841); 72 hr after transfection, GFP signal was measured following the manufacturer instructions (Ex. 505 Em. 515) with a Glomax reader.

## RNA isolation, RT-qPCR analysis, and RNA-sequencing

RNA was isolated using phenol-chloroform according to the manufacturer's instructions (TriPure Isolation Reagent, Sigma-Aldrich). RNA integrity was checked on the Agilent TapeStation using the RNA Screen Tape kit (Agilent). Total RNA (500 ng) was used to calculate mRNA levels by reverse transcription (RT) followed by quantitative (q)PCR analysis. RT was performed by using the Maxima Reverse Transcriptase protocol (Thermo Fisher) and qPCR analysis was carried out using specific primer pairs and SYBR green master mix (Kapa Biosystems) with a QuantStudio 5 Real-Time PCR System (Thermo Fisher). The sequences of primer oligos (from IDT) are listed in **Appendix-Key Resources Table**. Relative RNA levels were calculated by normalizing to *ACTB* mRNA, encoding β-Actin, or *18S* rRNA, using the $2^{-\Delta\Delta Ct}$ method. The specific normalization transcripts used in each case are indicated in the figure legends.

Sequencing libraries were prepared with TruSeq Stranded mRNA kit (Illumina) following the manufacturer's instructions. Final libraries were analyzed on the Agilent TapeStation using the D1000 Screen Tape kit and libraries were sequenced on an Illumina NovaSeq 6000 instrument with 250 cycles (paired-end, dual indexing). The RNA-seq reads were aligned to human genome hg19 Ensembl v82 using Spliced Transcripts Alignment to a Reference (STAR) software v2.4.0j and FeatureCounts (v1.6.4) to create gene counts. Differential gene expression analysis was performed with the DESeq2 package version 1.32.0 in R (v 4.1.0). The Wald test was used for statistical testing and mRNAs with Benjamini-Hochberg adjusted p-values <0.05 were used for further analysis with ShinyGO, EnrichR and MSigDB where indicated. RNA-seq datasets were deposited in GEO (GSE213993). Scripts are provided in *Source code 1*.

## SA-β-Gal activity

SA-β-Gal enzymatic activity assay was performed following the manufacturer's instructions (Cell Signaling Technology). Briefly, adherent cells were washed twice with 1×PBS, fixed for 15 min at 25 °C in the dark, and stained in an SA-β-Gal detection solution (pH 6.0) freshly prepared following the protocol provided by the manufacturer. Suspension cells (THP-1) were first seeded for 6 hr on poly-D-lysine (Gibco)-coated wells before starting the assay. Pictures were acquired by using a digital camera system (Nikon Digital Sight) adapted to a microscope (Nikon Eclipse TS100). SA-β-Gal activity was manually quantified by calculating the percentage of stained cells in three different fields per independent replicate. SPiDER-β-Gal activity was evaluated with the Senescence Cell Detection Kit (Dojindo, cat. SG05-01), following the manufacturer's instructions.

## Protein extraction, western blot, proteomics, and surface protein biotinylation and pulldown

To extract total protein, cells were washed twice in cold 1×PBS and harvested in 2% SDS in 50 mM HEPES buffer with freshly added protease and phosphatase inhibitors (Roche). Cell lysates were sonicated and centrifuged for 15 min at 12000×*g* to remove the insoluble fraction. Nuclear and cytoplasmic fractions were prepared using the NE-PER kit (Pierce) following the manufacturer's instructions. The protein content of the cleared lysates was quantified with the BCA assay (Pierce). Lysates were mixed with 4×SDS Laemmli buffer (Bio-Rad) and boiled at 95 °C for 5 min. For electrophoresis through SDS-containing polyacrylamide gels (SDS-PAGE) and western blot analysis, samples were loaded on

4–20% Tris-Glycine gels (Bio-Rad) and transferred onto nitrocellulose membranes using the iBlot kit (Invitrogen). Membranes were blocked for 1 hr at 25 °C with BSA or milk and incubated overnight with primary antibodies at 4 °C. A list of the antibodies used in this study is provided (**Appendix-Key Resources Table**). After washing with 1×TBST, the membranes were incubated with the secondary antibodies for 1 hr at 25 °C in 5% nonfat milk. After washes, the membranes were incubated with ECL solution (Kindle Biosciences) before acquiring chemiluminescent signals with a ChemiDoc system (Bio-Rad). Densitometry analysis was performed with ImageJ 1.52 A.

To prepare samples for proteomic analysis, $10^7$ THP-1 cells per sample were centrifuged for 5 min at 1000 x $g$ and washed twice in 1×PBS. Samples were shipped in dry ice for proteomic MS analysis (Poochon Scientific).

Surface proteins were isolated using the Cell Surface Protein Isolation Kit (Pierce, A44390) following the manufacturer's protocol. Briefly, WI-38 cells were washed with 1×PBS and labeled with a membrane-impermeant Sulfo-NHS-SS-biotin conjugate. Biotinylated proteins were captured on a neutravidin resin, washed and eluted in DTT. The eluted proteins were mixed with sample buffer and prepared for western blot analysis.

## NF-κB activity assay

NF-κB DNA-binding activity was measured using the TransAM NF-κB activity assay (Active Motif). Briefly, 2 µg of nuclear extract containing the activated transcription factors were added to a well coated with a consensus oligonucleotide for NF-κB binding. Samples were incubated for 1 hr at 25 °C, followed by incubation with primary antibodies that recognized the individual NF-κB subunits (p65, p50, p52, c-Rel, RelB). After incubation with secondary antibodies and signal development, absorbance was read on a microplate reader (Glomax, Promega) at 450 nm. For each different antibody, the absorbance was normalized to a specific blank control (lysis buffer instead of samples).

## ELISA and Luminex assay

The level of BAFF secreted in the cell culture media was assessed with the BAFF hypersensitive soluble human BAFF ELISA kit (sensitivity >8 pg/ml; Adipogen), following the manufacturer's instructions, and the plates were read on a Glomax microplate reader (Promega) at 450 nm. The levels of other cytokines and chemokines, as well as the level of cytokines in mouse serum, were assessed by using customized plates (Luminex assay, R&D) and analyzed on a Bio-Plex 200 instrument (Bio-Rad).

## Mice and doxorubicin-induced senescence in vivo

The experimental procedures related to animal work (ASP #474-LGG-2023) were approved by the Animal Care and Use Committee of the National Institute on Aging (NIA/NIH). Mice were imported from The Jackson Laboratory (Bar Harbor, ME) and housed in the animal facility at NIA. C57BL/6 mice at 10–12 weeks of age (all females) were treated systemically with doxorubicin to induce cellular senescence in tissues and organs in vivo. Briefly, a single dose of doxorubicin (10 mg/kg) and/or vehicle (DMSO) was injected intraperitoneally, and different tissues were collected 14 days later.

## Flow cytometry

THP-1 cells were pelleted by gentle centrifugation at 500×$g$ for 5 min, followed by washing and resuspension in 1×PBS. Cells were incubated with Fixable Viability Dye eFluor 780 (ThermoFisher, cat. 65-0865-14) for 30 min at 4 °C in the dark, washed with 1×PBS and analyzed (Ex. 633, Em. 780). To stain for cell-surface receptors, THP-1 cells were resuspended in PBS and incubated with antibodies directly conjugated to fluorescent dyes (anti-human BAFFR conjugated to FITC, BioLegend, cat. 316904; anti-human BCMA conjugated to APC, BioLegend, cat. 357505; anti-human TACI, conjugated to PE, cat. 311906, BioLegend) for 15 min at 20 °C in the dark. Following washes with 1×PBS, flow cytometry analysis was performed on a FACS Canto II flow cytometer (BD Biosciences).

To assess cell cycle distribution, THP-1 cells were incubated with Hoechst 33342 (ThermoFisher, cat. R37165) in DMEM for 30 min at 37 °C, harvested and then washed once with 1×PBS. Flow cytometry analysis was performed on a FACS Canto II flow cytometer (BD Biosciences), and data analysis was carried out using FlowJo software (BD Biosciences).

## Chromatin immunoprecipitation (ChIP)

Each ChIP reaction was performed with about $3 \times 10^5$ THP-1 cells, with the use of the MAGnify Chromatin Immunoprecipitation System Kit (ThermoFisher). Briefly, cells were crosslinked with formaldehyde for

10 min at room temperature, followed by cell lysis and chromatin isolation. Sonicated chromatin (200–500 bp) was immunoprecipitated for 4 hr at 4°C with 1 µg anti-IRF1 antibody (CST #8478) or rabbit IgG (CST #3900) conjugated to Dynabeads. The crosslinking was reversed by heat treatment, and the DNA associated with IRF1 was purified and used for downstream qPCR analysis. Primer sequences are listed in **Appendix-Key resources Table**.

### Cytokine array and phosphoarray

Cytokines and chemokines secreted in the culture media were analyzed with the Proteome Profiler Human Cytokine Array Kit (R&D) following the manufacturer's protocol. The relative levels of kinase phosphorylation in cell lysates were analyzed with the Proteome Profiler Human Phospho-Kinase Array Kit (R&D) following the manufacturer's instructions. Chemiluminescent signals from the cytokine array or phosphoarray were acquired on a ChemiDoc machine (Bio-Rad), and densitometry analysis was performed with ImageJ, version 1.52 A (NIH).

### Statistical analysis and graphs

Experiments were carried out three times unless otherwise stated. Data were tested for normal distribution and were compared by unpaired Student's $t$-test, using GraphPad Prism 9. Statistical significance was indicated as follows: *, $p<0.05$; **, $p<0.01$; ***, $p<0.001$; ****, $p<0.0001$. Graphs were generated using GraphPad Prism 9.

## Acknowledgements

We thank Dr. Nikki Noren Hooten and Dr. Michele K Evans (NIA/NIH) for suggestions throughout this research and for samples.

This work was funded by the National Institute on Aging Intramural Research Program of the National Institutes of Health. This research was also supported in part by the Intramural Research Program of the NIH, National Cancer Institute, Center for Cancer Research. The content of this publication does not necessarily reflect the views or policies of the Department of Health and Human Services, nor does mention of trade names, commercial products, or organizations imply endorsement by the U.S. Government.

## Additional information

### Funding

| Funder | Grant reference number | Author |
|---|---|---|
| National Institute on Aging | Z01-AG00511 | Martina Rossi |

The funders had no role in study design, data collection and interpretation, or the decision to submit the work for publication.

### Author contributions

Martina Rossi, Conceptualization, Data curation, Formal analysis, Investigation, Methodology, Writing – original draft, Writing – review and editing; Carlos Anerillas, Formal analysis, Investigation, Methodology, Writing – original draft; Maria Laura Idda, Conceptualization, Supervision, Investigation; Rachel Munk, Formal analysis, Validation, Investigation; Chang Hoon Shin, Xiaoling Yang, Resources, Investigation; Stefano Donega, Formal analysis, Visualization, Writing – original draft; Dimitrios Tsitsipatis, Conceptualization, Formal analysis, Investigation, Methodology; Allison B Herman, Conceptualization, Methodology; Jennifer L Martindale, Resources, Formal analysis, Investigation; Yulan Piao, Resources, Investigation, Methodology; Krystyna Mazan-Mamczarz, Resources, Software, Investigation, Methodology; Jinshui Fan, Investigation, Methodology; Luigi Ferrucci, Supervision, Funding acquisition; Peter F Johnson, Resources; Supriyo De, Conceptualization, Data curation, Software, Supervision, Project administration; Kotb Abdelmohsen, Conceptualization, Formal analysis, Supervision, Investigation, Writing – original draft; Myriam Gorospe, Supervision, Funding acquisition, Project administration, Writing – review and editing

### Author ORCIDs
Martina Rossi http://orcid.org/0000-0001-7738-9841
Luigi Ferrucci http://orcid.org/0000-0002-6273-1613
Myriam Gorospe http://orcid.org/0000-0001-5439-3434

### Ethics
"All mouse work, including the import, housing, experimental procedures, and euthanasia, was done under an Animal Study Proposal (ASP #474-LGG-2023) that was reviewed and approved by the Animal Care and Use Committee (ACUC) of the National Institute on Aging (NIA), NIH."

### Decision letter and Author response
Decision letter https://doi.org/10.7554/eLife.84238.sa1
Author response https://doi.org/10.7554/eLife.84238.sa2

---

## Additional files

### Supplementary files
• MDAR checklist
• Source code 1. Alignment, gene count and differential gene expression scripts.

### Data availability
RNA-seq data are deposited in GSE213993. Proteomic data are in Source Data 1.

The following dataset was generated:

| Author(s) | Year | Dataset title | Dataset URL | Database and Identifier |
|---|---|---|---|---|
| Rossi M | 2023 | Pleiotropic effects of BAFF on the senescence-associated secretome and growth arrest | http://www.ncbi.nlm.nih.gov/geo/query/acc.cgi?acc=GSE213993 | NCBI Gene Expression Omnibus, GSE213993 |

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

# Appendix 1

**Appendix 1—key resources table**

| Reagent type (species) or resource | Designation | Source or reference | Identifiers | Additional information |
|---|---|---|---|---|
| Gene (*Homo sapiens*) | *TNFSF13B, (BAFF)* | Ensembl | ENSG00000 102524 | TNF superfamily member 13b |
| Gene (*Mus musculus*) | *Tnfsf13b, (Baff)* | Ensembl | ENSMUSG0000 0031497 | TNF superfamily member 13b |
| Gene (*Homo sapiens*) | *IRF1* | Ensembl | ENSG00000 125347 | Interferon regulatory factor 1 |
| Cell line (*Homo sapiens*) | THP-1 | ATCC | TIB-202 | Cell line Monocytes (AML) Sex: male |
| Cell line (*Mus musculus*) | RAW 264.7 | ATCC | TIB-71 | Cell line Macrophages/monocytes Karyotype: unspecified Sex: male |
| Cell line (*Homo sapiens*) | WI-38 | Coriell Institute | AG06814-J | primary fibroblasts Karyotype: normal, diploid Sex: female |
| Cell line (*Homo sapiens*) | IMR-90 | Coriell Institute | I90-79 | primary fibroblasts Karyotype: normal, diploid Sex: female |
| Antibody (mouse, monoclonal) | Actin | SCBT | sc-8432 | 1:10000 |
| Antibody (mouse, monoclonal) | GAPDH | SCBT | sc-47724 | 1:2000 |
| Antibody (mouse, monoclonal) | Tubulin | SCBT | sc-5286 | 1:2000 |
| Antibody (rabbit, monoclonal) | BAFF (human) | CST | 19944 | 1:500 |
| Antibody (rabbit, monoclonal) | BAFF (human) | Abcam | ab224710 | 1:500 |
| Antibody (rabbit, polyclonal) | BAFF (mouse) | Millipore Sigma | AB16530 | 1:500 |
| Antibody (rabbit, polyclonal) | IRF1 | ABClonal | A7692 | 1:500 |
| Antibody (rabbit, monoclonal) | IRF1 | CST | 8478 | Used for ChIP 1 μg / reaction |
| Antibody (rabbit, polyclonal) | IRF2 | ABClonal | A4843 | 1:500 |
| Antibody (rabbit, monoclonal) | p21 | CST | 2947 | 1:500 |
| Antibody (rabbit, monoclonal) | DPP4 | CST | 67138 | 1:1000 |
| Antibody (mouse, monoclonal) | p53-DO1 (human) | SCBT | sc-126 | 1:2000 |
| Antibody (mouse monoclonal) | p53-Pab240 (mouse) | Abcam | Ab26 | 1:1000 |
| Antibody (rabbit monoclonal) | p53-Ser15 (mouse) | CST | 12571 | 1:500 |

*Appendix 1 Continued on next page*

*Appendix 1 Continued*

| Reagent type (species) or resource | Designation | Source or reference | Identifiers | Additional information |
|---|---|---|---|---|
| Antibody | PARP1 | CST | 9425 | 1:2000 |
| Antibody (mouse monoclonal) | Cathepsin B | SCBT | sc-365558 | 1:1000 |
| Antibody (rabbit, monoclonal) | RAB1A | CST | 13075 | 1:1000 |
| Antibody (mouse monoclonal) | NCF1 | SCBT | sc-17845 | 1:1000 |
| Antibody (mouse monoclonal) | BAFFR | SCBT | sc-365410 | 1:500 |
| Antibody (rabbit, monoclonal) | TACI | CST | 96641 | 1:500 |
| Antibody (rabbit, monoclonal) | BCMA | CST | 88183 | 1:500 |
| Antibody (rabbit, monoclonal) | p65 | CST | 8242 | 1:1000 |
| Antibody (rabbit, monoclonal) | Ac-p65-K310 | CST | 12629 | 1:500 |
| Antibody (mouse, monoclonal) | Ikba | SCBT | sc-1643 | 1:500 |
| Antibody (rabbit, polyclonal) | p50 | CST | 3035 | 1:1000 |
| Antibody (rabbit, polyclonal) | p52 | CST | 4882 | 1:1000 |
| Recombinant DNA reagent (human) | *TNFSF13B* ORF Clone | GenScript | OHu22261 | BAFF overexpression plasmid |
| Sequence-based reagent | *TNFSF13B f* | IDT | PCR primers | CACAATTCAAAGGGGCAGTAA |
| Sequence-based reagent | *TNFSF13B r* | IDT | PCR primers | ACTGAAAAGGAGGGAGTGCAT |
| Sequence-based reagent | *TNFSF13B pre-mrna f* | IDT | PCR primers | CTGGAAGAGTGGGTTTCTAGC |
| Sequence-based reagent | *TNFSF13B pre-mRNA r* | IDT | PCR primers | GTTGGTGTTTCACTGTCTGCAATC |
| Sequence-based reagent | *CDKN1A f* | IDT | PCR primers | AGTCAGTTCCTTGTGGAGCC |
| Sequence-based reagent | *CDKN1A r* | IDT | PCR primers | CATGGGTTCTGACGGACAT |
| Sequence-based reagent | *IRF1 f* | IDT | PCR primers | CATGCCCTCCACCTCTGAAG |
| Sequence-based reagent | *IRF1r* | IDT | PCR primers | CCATCCACGTTTGTTGGCTG |
| Sequence-based reagent | *IRF2 f* | IDT | PCR primers | TCCTGAGTATGCGGTCCTGA |
| Sequence-based reagent | *IRF2 r* | IDT | PCR primers | AGATGGGACTGTCCTACAACT |
| Sequence-based reagent | *IRF8 f* | IDT | PCR primers | AGGTCTTCGACACCAGCCAGTT |

*Appendix 1 Continued on next page*

*Appendix 1 Continued*

| Reagent type (species) or resource | Designation | Source or reference | Identifiers | Additional information |
|---|---|---|---|---|
| Sequence-based reagent | IRF8 r | IDT | PCR primers | GCACGAGAATGAGTTTGGAGCG |
| Sequence-based reagent | BLIMP1 f | IDT | PCR primers | CAGTTCCTAAGAACGCCAACAGG |
| Sequence-based reagent | BLIMP1 r | IDT | PCR primers | GTGCTGGATTCACATAGCGCATC |
| Sequence-based reagent | MYC f | IDT | PCR primers | TTCTCTCCGTCCTCGGATTCTCTG |
| Sequence-based reagent | MYC r | IDT | PCR primers | TCTTCTTGTTCCTCCTCAGAGTCG |
| Sequence-based reagent | TNFSF13B promoter f | IDT | PCR primers | AGCAGACAGAGTTCCCTTGC |
| Sequence-based reagent | TNFSF13B promoter r | IDT | PCR primers | TGGAGTTTGGATTGGCACAG |
| Sequence-based reagent | GAPDH promoter f | IDT | PCR primers | GCCAATCTCAGTCCCTTCCC |
| Sequence-based reagent | GAPDH promoter r | IDT | PCR primers | TAGTAGCCGGGCCCTACTTT |
| Sequence-based reagent | MX1 promoter f | IDT | PCR primers | CACTGCCCCCTCGTCGTGGCACCGC |
| Sequence-based reagent | MX1 promoter r | IDT | PCR primers | TTTCTGCTCGCTGGTTTCCAGA |
| Sequence-based reagent | FUCA1 f | IDT | PCR primers | GACTTCGGACCGCAGTTCACTG |
| Sequence-based reagent | FUCA 1 r | IDT | PCR primers | CCAGTTCCAAGACACAGGACTC |
| Sequence-based reagent | S100A9 f | IDT | PCR primers | GCACCCAGACACCCTGAACCA |
| Sequence-based reagent | S100A9 r | IDT | PCR primers | TGTGTCCAGGTCCTCCATGATG |
| Sequence-based reagent | CD14 f | IDT | PCR primers | CTGGAACAGGTGCCTAAAGGAC |
| Sequence-based reagent | CD14 r | IDT | PCR primers | GTCCAGTGTCAGGTTATCCACC |
| Sequence-based reagent | PTGS2 f | IDT | PCR primers | CGGTGAAACTCTGGCTAGACAG |
| Sequence-based reagent | PTGS2 r | IDT | PCR primers | GCAAACCGTAGATGCTCAGGGA |
| Sequence-based reagent | CTSB f | IDT | PCR primers | GCTTCGATGCACGGGAACAATG |
| Sequence-based reagent | CTSB r | IDT | PCR primers | CATTGGTGTGGATGCAGATCCG |
| Sequence-based reagent | CXCL1 f | IDT | PCR primers | AGCTTGCCTCAATCCTGCATCC |
| Sequence-based reagent | CXCL1 r | IDT | PCR primers | TCCTTCAGGAACAGCCACCAGT |
| Sequence-based reagent | S100A8 f | IDT | PCR primers | ATGCCGTCTACAGGGATGACCT |

*Appendix 1 Continued on next page*

*Appendix 1 Continued*

| Reagent type (species) or resource | Designation | Source or reference | Identifiers | Additional information |
|---|---|---|---|---|
| Sequence-based reagent | S100A8 r | IDT | PCR primers | AGAATGAGGAACTCCTGGAAGTTA |
| Sequence-based reagent | DUSP1 f | IDT | PCR primers | CAACCACAAGGCAGACATCAGC |
| Sequence-based reagent | DUSP1 r | IDT | PCR primers | GTAAGCAAGGCAGATGGTGGCT |
| Sequence-based reagent | MMP9 f | IDT | PCR primers | GCCACTACTGTGCCTTTGAGTC |
| Sequence-based reagent | MMP9 r | IDT | PCR primers | CCCTCAGAGAATCGCCAGTACT |
| Sequence-based reagent | HMOX1 f | IDT | PCR primers | CCAGGCAGAGAATGCTGAGTTC |
| Sequence-based reagent | HMOX1 r | IDT | PCR primers | AAGACTGGGCTCTCCTTGTTGC |
| Sequence-based reagent | CCL2 f | IDT | PCR primers | AGAATCACCAGCAGCAAGTGTCC |
| Sequence-based reagent | CCL2 r | IDT | PCR primers | TCCTGAACCCACTTCTGCTTGG |
| Sequence-based reagent | IL6 f | IDT | PCR primers | AGTGAGGAACAAGCCAGAGC |
| Sequence-based reagent | IL6 r | IDT | PCR primers | GTCAGGGGTGGTTATTGCAT |
| Sequence-based reagent | IL8 f | IDT | PCR primers | TCCTGATTTCTGCAGCTCTGT |
| Sequence-based reagent | IL8 r | IDT | PCR primers | AAATTTGGGGTGGAAAGGTT |
| Sequence-based reagent | IL1B f | IDT | PCR primers | TCCAGGGACAGGATATGGAG |
| Sequence-based reagent | IL1B r | IDT | PCR primers | CCAAGGCCACAGGTATTTTG |
| Sequence-based reagent | SERPINE1 f | IDT | PCR primers | CTCATCAGCCACTGGAAAGGCA |
| Sequence-based reagent | SERPINE1 r | IDT | PCR primers | GACTCGTGAAGTCAGCCTGAAAC |
| Sequence-based reagent | ICAM1 f | IDT | PCR primers | AGCGGCTGACGTGTGCAGTAAT |
| Sequence-based reagent | ICAM1 r | IDT | PCR primers | TCTGAGACCTCTGGCTTCGTCA |
| Sequence-based reagent | CCL5 f | IDT | PCR primers | CCTGCTGCTTTGCCTACATTGC |
| Sequence-based reagent | CCL5 r | IDT | PCR primers | ACACACTTGGCGGTTCTTTCGG |
| Sequence-based reagent | TNFA f | IDT | PCR primers | AGAACTCACTGGGGCCTACA |
| Sequence-based reagent | TNFA r | IDT | PCR primers | AGGAAGGCCTAAGGTCCACT |
| Sequence-based reagent | BAFFR f | IDT | PCR primers | GTCTCCGGGAATCTCTGATGC |

*Appendix 1 Continued on next page*

*Appendix 1 Continued*

| Reagent type (species) or resource | Designation | Source or reference | Identifiers | Additional information |
|---|---|---|---|---|
| Sequence-based reagent | *BAFFR r* | IDT | PCR primers | GTTCAGTGGAGCCCAGCTCT |
| Sequence-based reagent | *TACI f* | IDT | PCR primers | CTGTGGACAGCACCCTAAGC |
| Sequence-based reagent | *TACI r* | IDT | PCR primers | CAACTTCTCCACTCCGCTGTC |
| Sequence-based reagent | *BCMA f* | IDT | PCR primers | GGGCAGTGCTCCCAAAATGA |
| Sequence-based reagent | *BCMA r* | IDT | PCR primers | AACGCTGACATGTTAGAGGAGG |
| Sequence-based reagent | *CYFIP2 f* | IDT | PCR primers | ATGCCCTGGATTCTAACGGACC |
| Sequence-based reagent | *CYFIP2 r* | IDT | PCR primers | CTTGGTCAGAGCATAGTAGGCG |
| Sequence-based reagent | *CLCA2 f* | IDT | PCR primers | CCAGACTGCCAAGGAATCCATTG |
| Sequence-based reagent | *CLCA2 r* | IDT | PCR primers | GCTTCTGTGATTGCACATTTTGTTC |
| Sequence-based reagent | *LIF f* | IDT | PCR primers | CCAACGTGACGGACTTCCC |
| Sequence-based reagent | *LIF r* | IDT | PCR primers | TACACGACTATGCGGTACAGC |
| Sequence-based reagent | *AURKB f* | IDT | PCR primers | GGAGTGCTTTGCTATGAGCTGC |
| Sequence-based reagent | *AURKB r* | IDT | PCR primers | GAGCAGTTTGGAGATGAGGTCC |
| Sequence-based reagent | *CDK1 f* | IDT | PCR primers | GGAAACCAGGAAGCCTAGCATC |
| Sequence-based reagent | *CDK1 r* | IDT | PCR primers | GGATGATTCAGTGCCATTTTGCC |
| Sequence-based reagent | *KI-67 f* | IDT | PCR primers | GAAAGAGTGGCAACCTGCCTTC |
| Sequence-based reagent | *KI-67 r* | IDT | PCR primers | GCACCAAGTTTTACTACATCTGCC |
| Sequence-based reagent | *LMNB1 f* | IDT | PCR primers | GAGAGCAACATGATGCCCAAGTG |
| Sequence-based reagent | *LMNB1 r* | IDT | PCR primers | GTTCTTCCCTGGCACTGTTGAC |
| Sequence-based reagent | *LBR f* | IDT | PCR primers | CTATGTGGTGGATGCTCTCTGG |
| Sequence-based reagent | *LBR r* | IDT | PCR primers | CCACACCAAGTCTCCAAAAGCC |
| Sequence-based reagent | *ACTB f* | IDT | PCR primers | GCACAGAGCCTCGCCTT |
| Sequence-based reagent | *ACTB r* | IDT | PCR primers | GTTGTCGACGACGAGCG |
| Sequence-based reagent | *18 S f* | IDT | PCR primers | ACCCGTTGAACCCCATTCGTGA |

*Appendix 1 Continued on next page*

*Appendix 1 Continued*

| Reagent type (species) or resource | Designation | Source or reference | Identifiers | Additional information |
|---|---|---|---|---|
| Sequence-based reagent | *18 S r* | IDT | PCR primers | GCCTCACTAAACCATCCAATCGG |
| Sequence-based reagent | *GAPDH f* | IDT | PCR primers | CTCTGCTCCTCCTGTTCGAC |
| Sequence-based reagent | *GAPDH r* | IDT | PCR primers | ACGACCAAATCCGTTGACTC |
| Sequence-based reagent | *mTnfsf13b f* | IDT | PCR primers | CTACCGAGGTTCAGCAACACCA |
| Sequence-based reagent | *mTnfsf13b r* | IDT | PCR primers | GAAAGCGCGTCTGTTCCTGTGG |
| Sequence-based reagent | *mCdkn1a f* | IDT | PCR primers | TTGCCAGCAGAATAAAAGGTG |
| Sequence-based reagent | *mCdkn1a r* | IDT | PCR primers | TTTGCTCCTGTGCGGAAC |
| Sequence-based reagent | *m18S f* | IDT | PCR primers | AGTCCCTGCCCTTTGTACACA |
| Sequence-based reagent | *m18S r* | IDT | PCR primers | CGATCCGAGGGCCTCACTA |
| Sequence-based reagent | *mActb f* | IDT | PCR primers | TTCTTTGCAGCTCCTTCGTT |
| Sequence-based reagent | *mActb r* | IDT | PCR primers | ATGGAGGGGAATACAGCCC |
| Sequence-based reagent | Control siRNA-A | SCBT | sc-37007 | |
| Sequence-based reagent | IRF1 siRNA | SCBT | sc-35706 | |
| Sequence-based reagent | Non targeting pool | Horizon Discovery | Cat. D-001206-14-20 | |
| Sequence-based reagent | IRF1 siRNA | Horizon Discovery | M-011704-01-0005 | |
| Sequence-based reagent | BAFF siRNA | Horizon Discovery | D-017586-01-0005 | |
| Sequence-based reagent | BAFF siRNA | Horizon Discovery | D-017586-03-0005 | |
| Sequence-based reagent | BAFF siRNA | Horizon Discovery | M-046829-00-0010 | |
| Sequence-based reagent | BAFF-R siRNA | Horizon Discovery | M-013424-00-0005 | |
| Sequence-based reagent | BCMA siRNA | Horizon Discovery | M-011217-02-0005 | |
| Sequence-based reagent | TACI siRNA | Horizon Discovery | M-008095-00-0005 | |
| Commercial assay or Kit | Senescence β-Galactosidase Staining Kit | Cell Signaling | #9860 | Senescent cells detection kit |
| Commercial assay or Kit | Luminex Discovery Assay | R&D System | LXSAHM | Custom-designed plates for multiplex ELISA |

*Appendix 1 Continued on next page*

*Appendix 1 Continued*

| Reagent type (species) or resource | Designation | Source or reference | Identifiers | Additional information |
|---|---|---|---|---|
| Commercial assay or Kit | Amaxa Nucleofector kit V | Lonza | VCA-1003 | Electroporation of overexpression plasmids into THP-1, program V-001 |
| Commercial assay or Kit | Magnify ChIP System | Thermofisher | 492024 | |
| Commercial assay or Kit | TransAM NFKB Family | Active Motif | 43296 | |
| Commercial assay or Kit | Human Phospho-Kinase Array Kit | R&D | ARY003C | |
| Commercial assay or Kit | Cytokine array kit | R&D | ARY005B | |
| Chemical compound, drug | Ruxolitinib | Invivogen | tlrl-rux | |
| Chemical compound, drug | MRT67307 | Invivogen | inh-mrt | |
| Chemical compound, durg | BX-795 | Invivogen | tlrl-bx7 | |
| Software, algorithm | GraphPad | Prism | (https://graphpad.com) | |
| Software, algorithm | ShinyGO | South Dakota State University | http://bioinformatics.sdstate.edu/go/ | |

