## [Editor Report]

Rossi et al. carry out a valuable characterization of the molecular circuitry connecting the immunomodulatory cytokine BAFF (B-cell activating factor) in the context of cellular senescence. They present solid evidence that BAFF is upregulated in response to senescence, and that this upregulation is partially driven by the immune response-regulating transcription factor (TF) IRF1, with potential cell type-specific effects during senescence. Ultimately, these results strongly suggest that BAFF plays a senomorphic role in senescence, modulating downstream senescence-associated phenotypes, and may be an interesting candidate for senomorphic therapy.

---

## [Decision Letter]

**Decision letter after peer review:**

Thank you for submitting your article "Pleiotropic effects of BAFF on the senescence-associated secretome and growth arrest" for consideration by *eLife*. Your article has been reviewed by 3 peer reviewers, one of whom is a member of our Board of Reviewing Editors, and the evaluation has been overseen by Satyajit Rath as the Senior Editor. The reviewers have opted to remain anonymous.

Essential revisions:

The reviewers discussed the manuscript, and believe the study is interesting but has some concerns of a technical and methodological nature that they believe need to be addressed. Specifically:

1. Some specificity and technical controls are currently missing:

– Technical controls: validating siRNA specificity/efficiency by Western, antibody specificity, consistency between panels.

– Biological control: changes in aging tissues, specificity to senescence vs. quiescence, degree of senescence induction, cell type specificity vs. primary/cancer specificity (i.e. also include primary monocytes).

2. There also are some methodological issues to be cleared up to ensure data robustness.

– See comments on issues with the RNAseq analysis pipeline as currently implemented

– See comments on the need to use at least 2 independent siRNA to exclude that observed phenotypes may be due to off-target effects.

– See comments on the need to use 2-3 RT-qPCR normalizing amplicons to avoid noise due to housekeeping genes not being constant across conditions

*Reviewer #1 (Recommendations for the authors):*

1. The model proposed by the authors, as currently illustrated in Figure 7, is too general to be supported by the data. While this reviewer agrees that the contributions of BAFF to senescence responses are likely to be cell-type specific, it is unclear that p53 and NF-κB actually play differential roles in senescent fibroblasts and senescent monocytes. For the series of monocyte experiments, cancerous THP-1 cells were used which, as stated by the authors, harbor mutations in p53 that prevent its expression. Thus, it is unclear whether the p53 response is specific to fibroblasts, or whether the lack of a p53 response is an artifact/feature of this specific cancer monocytic cell line. That is, it is still within the realm of possibility that the p53 response may be influenced by BAFF in non-cancerous, senescent monocytes with intact p53. This remains an open question. Since the authors have access to mice based on some of the analyses, a relatively easy source of monocytes with intact p53 can be from bone marrow, which would help strengthen the point they make on the impact of NF-κB. The model, and its discussion, should be revised to take this caveat into account (or additional experiments in WT/primary cells should be added).

2. Currently, RT-qPCR data is normalized to a single reference gene (ACTB or GAPDH), e.g. in Figure 1G. However, results using a single reference gene are often unstable, since no gene is truly ever impacted by outside stimuli. It is considered a best practice to include multiple reference genes when conducting this type of analysis [PMID: 19246619], for instance using the geometric mean of the Cts of 3+ normalizing amplicons. Although control using 18s rRNA is mentioned in Figure legend, it does not seem like it was used to perform normalization of the RT-qPCR results shown in Figure 1G.

2.1. Importantly, the cytoskeleton is known to be broadly impacted by senescence [PMID: 15742196], which makes ACTB a poor choice for a normalizing amplicon. Consistent with using ACTB as the single "housekeeping" gene being an issue, ACTB protein levels are highly variable in the spleens of Dox-treated mice in Figure 1H, suggesting that mRNA levels may also be highly variable. Thus, presented RT-qPCR data should be revised or re-run accordingly using a panel of normalizing amplicons (unless RNA-seq has been performed for the same/similar samples and the RNA-seq results can be shown as well for discussed genes, as RNA-seq normalization bypasses housekeeping gene-related issues).

2.2. In addition, if different normalizing amplicons are used for different panels, please make sure to note which were used for which, as well as a rationale for using different ones for different analyses.

3. The authors state that DEGs were defined as those genes with padj < 0.05 and absolute log2 fold change > 1. However, it is unclear how fold change filtering was implemented. Fold-change filtering, if not implemented into the statistical model used to identify DEGs, can lead to poor FDR control and is not appropriate (see PMC2654802). Wald tests in DESeq2 can be constructed with thresholds; please run the analysis in this manner if fold change filtering is currently carried out post hoc using the default settings. Alternatively, DESeq's default settings can be used without fold change filtering – if the number of DEGs is the issue, a smaller FDR threshold can be used (e.g. FDR < 0.01). Additionally, please ensure that DEGs are called uniformly throughout the manuscript (or explicitly justify differences between panels). In contrast to the methods section, Figure 3B implements an abs (fold change) > 1.3 thresholds.

4. Experimental groups were statistically analyzed using unpaired Student's t-tests. Though the data were tested for normality, it is difficult to say whether a sample size of 3 actually came from a normal distribution. To avoid making normality assumptions, analyses should be carried out using non-parametric tests instead, such as Wilcoxon tests. Most effects identified in the manuscript will likely be robust enough to this change.

5. For reproducibility of code and analyses, all analytical scripts for this study should be deposited in a repository such as GitHub or made available as a Supplementary file.

*Reviewer #2 (Recommendations for the authors):*

While we believe, the manuscript as a whole is close to publication quality, one of the major concerns is the reliance on cell lines as the major model system used, as opposed to primary cells and more in-depth in vivo analysis to strengthen the relevance of BAFF in aging. Please see our comments and concerns below:

1) The amount of DNA damaging stimuli seems fairly weak (5gy IR and 10nM Doxo). Can the authors determine via cell cycle analysis or some other assay, what percent of cells are undergoing cell cycle arrest?

2) Likewise, in Figure 1 the SA-B-GAL staining in THP-1 cells seems to be only in a small subset of cells, can the authors also quantify the staining perhaps via the use of C12FDG using a plate reader or preferably flow cytometry?

3) Figure 1: In addition to doxo-treated mice, can the authors also measure BAFF expression in old vs young tissues?

4) Pg 6: "In all cases, BAFF mRNA levels increased during senescence (Figure 1—figure supplement 1E) and secreted BAFF was generally elevated with senescence, although it was undetectable in hVSMCs, and the levels were overall higher in THP-1 cells (Figure 1D and Figure 1—figure supplement 1F)." The total BAFF in the ELISA experiments in THP-1 and WI-38 cells is equivalent, in fact, the irradiated WI-38 cells have more expression than the irradiated THP-1 cells.

5) Pg 6: "Next, we investigated if the rise in BAFF mRNA with senescence in THP-1 cells was the result of transcriptional or posttranscriptional regulatory mechanisms" Please explain the rationale for asking this question.

6) Figure 2G: The authors show that IRF1 and IRF2 are induced during senescence and may drive BAFF expression. Thus, in addition to IRF1 can the authors also silence IRF2 to test its effect on BAFF mRNA expression.

7) In Figures 4B and 4C, can the cell viability and cell proliferation be measured via flow cytometry which is more quantitative?

8) Figure 5: To further define the BAFF signaling mechanism, can the authors also silence or KO the different BAFF receptors to determine which is the major receptor necessary for BAFF signaling in senescent THP-1 and fibroblast?

9) Figure 5: We suggest the authors perform an NFkB luciferase assay to confirm BAFF silencing impacts NFkB gene expression.

10) Pg 12 Figure 6: "To gain a more complete understanding of the role of BAFF in senescence, we investigated its function in primary fibroblasts, which are well-established models for senescence and express the senescence-relevant protein p53" The subsequent experiments were not performed in primary fibroblast.

11) Lastly, as mentioned above the paper would be strengthened if some of the key experiments can be performed in primary mouse or human monocytes and fibroblast to confirm BAFF regulates senescence in cells that are more relevant to normal physiology.

*Reviewer #3 (Recommendations for the authors):*

1. Figure 1. Test whether the upregulation of BAFF is specific to senescence, or also in reversible quiescence arrest.

2. Figure 1, Supplement 1G. Show negative control IgG for immunofluorescence.

3. All results with siRNA should be validated with at least 2 individual siRNAs to eliminate the possibility of off-target effects.

4. To confirm a role for IRF1 in the activation of BAFF, the authors should confirm the binding of IRF1 to the BAFF promoter by ChIP or ChIP-seq.

5. Key antibodies should be validated by siRNA knockdown of their targets, for example, TACI, BCMA, and BAFF-R in Figure 5. Note that there is an apparent discrepancy between BCMA data in Figure 5B vs 5C.

6. Figure 5E. Negative/specificity controls for this assay should be shown.

7. Hybridization arrays such as Figure 5H, Figure 6 – Supplement 1I, and Figure 6H should be shown as quantitated, normalized data with statistics from replicates.

8. Figure 6B – Supplement 1. Controls to confirm fractionation (i.e., non-contamination by cytosolic and nuclear proteins) should be shown.

9. Figure 6A. Knockdown of BAFF should be shown by western blot.

10. Figure 6G. Although BAFF knockdown decreases the expression of p53, p21 increases. How do the authors explain this?

---

## [Author Response]

Essential revisions:Reviewer #1 (Recommendations for the authors):1. The model proposed by the authors, as currently illustrated in Figure 7, is too general to be supported by the data. While this reviewer agrees that the contributions of BAFF to senescence responses are likely to be cell-type specific, it is unclear that p53 and NF-κB actually play differential roles in senescent fibroblasts and senescent monocytes. For the series of monocyte experiments, cancerous THP-1 cells were used which, as stated by the authors, harbor mutations in p53 that prevent its expression. Thus, it is unclear whether the p53 response is specific to fibroblasts, or whether the lack of a p53 response is an artifact/feature of this specific cancer monocytic cell line. That is, it is still within the realm of possibility that the p53 response may be influenced by BAFF in non-cancerous, senescent monocytes with intact p53. This remains an open question. Since the authors have access to mice based on some of the analyses, a relatively easy source of monocytes with intact p53 can be from bone marrow, which would help strengthen the point they make on the impact of NF-κB. The model, and its discussion, should be revised to take this caveat into account (or additional experiments in WT/primary cells should be added).

We thank the Reviewer for these insightful comments and we understand the concerns regarding the proposed model. To address the Reviewer’s requests, we performed further experiments to determine whether the p53 response is specific to fibroblasts or could be extended to p53-proficient cells of the monocyte/macrophage lineage. First, we followed the Reviewer’s suggestion to try to silence BAFF and induce senescence in primary CD14+ monocytes; unfortunately, due to their short lifespan in culture (2-4 days) and the fact that they are inherently difficult to transfect, this experiment was not technically possible. Instead, we investigated murine RAW 264.7 macrophages, which, despite being transformed, express p53 and are widely used to study p53 function in the immune system (PMID 25098341, PMID 16046548, PMID 29989329). Interestingly, despite the fact that total p53 levels remain unchanged after BAFF depletion, the levels of phospho-p53 (Ser15) in RAW 264.7 cells were reduced (new data in Figure 7 and Figure 7—figure supplement 1). These data suggest that BAFF regulates p53 not only in fibroblasts, but also in other p53-proficient cell types. Therefore, we modified our model in Figure 8 (previously Figure 7) to reflect the fact that p53 is implicated in this response in cells other than fibroblasts. We thank the Reviewer for helping us to improve this critical aspect of the manuscript.

2. Currently, RT-qPCR data is normalized to a single reference gene (ACTB or GAPDH), e.g. in Figure 1G. However, results using a single reference gene are often unstable, since no gene is truly ever impacted by outside stimuli. It is considered a best practice to include multiple reference genes when conducting this type of analysis [PMID: 19246619], for instance using the geometric mean of the Cts of 3+ normalizing amplicons. Although control using 18s rRNA is mentioned in Figure legend, it does not seem like it was used to perform normalization of the RT-qPCR results shown in Figure 1G.

We agree with the Reviewer’s comments. For Figure 1G, it is preferable to show the RT-qPCR data normalized to the levels of *18S* rRNA in order to avoid confusion, particularly as the levels of the protein ACTB (β-actin) protein sometimes fluctuate (Figure 1H). We have updated Figure 1G and we now specify in the legend that *18S* rRNA levels are used for normalization. We also note that in our RT-qPCR analysis, we routinely include two or three RNAs (e.g., *GAPDH* mRNA, *ACTB* mRNA, *18S* rRNA, etc) to ensure that our data are robust and consistent. For simplicity, in the manuscript we show only one reference RNA per panel.

2.1. Importantly, the cytoskeleton is known to be broadly impacted by senescence [PMID: 15742196], which makes ACTB a poor choice for a normalizing amplicon. Consistent with using ACTB as the single "housekeeping" gene being an issue, ACTB protein levels are highly variable in the spleens of Dox-treated mice in Figure 1H, suggesting that mRNA levels may also be highly variable. Thus, presented RT-qPCR data should be revised or re-run accordingly using a panel of normalizing amplicons (unless RNA-seq has been performed for the same/similar samples and the RNA-seq results can be shown as well for discussed genes, as RNA-seq normalization bypasses housekeeping gene-related issues).

We thank the Reviewer for suggesting a superior way to normalize our data. Based on our previous transcriptomic analysis using a range of models of cell senescence [Casella et al., *Nucleic Acids Res*, 2019 (PMID 31251810)], *ACTB* mRNA was not significantly altered upon senescence in WI-38 or IMR-90 fibroblasts. For THP-1 cells (which were not included in the aforementioned study by Casella et al.), we did re-analyze some key results by normalizing to *18S* rRNA levels. As shown in Author response image 1, normalization to *ACTB* mRNA levels (left) yielded quite similar results as normalization to *18S* rRNA levels (right). For example, the increase in *TNFSF13B* mRNA upon senescence and the effects of BAFF silencing on SASP factor mRNAs (*IL8*, *TNFA*, and *IL1B* mRNAs) were comparable whether we normalized the data to the levels of *18S* rRNA or *ACTB* mRNA. We chose *ACTB* mRNA as a normalization housekeeping transcript in the manuscript, but we provide the comparison data for the Reviewer.

**Author response image 1. sa2fig1:** 

2.2. In addition, if different normalizing amplicons are used for different panels, please make sure to note which were used for which, as well as a rationale for using different ones for different analyses.

The Reviewer makes several valuable points. We apologize for neglecting to indicate the normalizing amplicons used in each panel. We have updated our Figure Legends to reflect this important information. We used *ACTB* mRNA for normalization throughout the manuscript, except for Figure 1G (RT-qPCR analysis in mouse tissues), where we used *18S* rRNA for normalization, as suggested by the Reviewer (point 2), and Figure 2—figure supplement 1A,C,D, where we used *18S rRNA* for normalization*.* The reason for this choice is that mRNA stability experiments using actinomycin D (a drug that inhibits RNA polymerase II, which transcribes all mRNAs, including *TNFSF13B* and *ACTB* mRNAs) are designed to require normalization to *18S* rRNA levels. The reason behind this choice is that RNA polymerase I (not RNA polymerase II) transcribes the parent transcript of *18S* rRNA, and hence *18S* rRNA levels are refractory to inhibition by actinomycin D treatment.

3. The authors state that DEGs were defined as those genes with padj < 0.05 and absolute log2 fold change > 1. However, it is unclear how fold change filtering was implemented. Fold-change filtering, if not implemented into the statistical model used to identify DEGs, can lead to poor FDR control and is not appropriate (see PMC2654802). Wald tests in DESeq2 can be constructed with thresholds; please run the analysis in this manner if fold change filtering is currently carried out post hoc using the default settings. Alternatively, DESeq's default settings can be used without fold change filtering – if the number of DEGs is the issue, a smaller FDR threshold can be used (e.g. FDR < 0.01). Additionally, please ensure that DEGs are called uniformly throughout the manuscript (or explicitly justify differences between panels). In contrast to the methods section, Figure 3B implements an abs (fold change) > 1.3 thresholds.

We apologize for the confusion and for the inconsistencies between the Materials and methods section and the figures. Following the Reviewer’s suggestion, we re-analyzed our RNA-seq data from THP-1 and WI-38 cells without calculating fold change (we maintained the cutoff padj<0.05), and we updated the pertinent sections of the manuscript following this updated analysis.

4. Experimental groups were statistically analyzed using unpaired Student's t-tests. Though the data were tested for normality, it is difficult to say whether a sample size of 3 actually came from a normal distribution. To avoid making normality assumptions, analyses should be carried out using non-parametric tests instead, such as Wilcoxon tests. Most effects identified in the manuscript will likely be robust enough to this change.

We thank the reviewer for his valuable comment. Given the small sample size (<7), however, the Wilcoxon Rank Sum Test (or Mann-Whitney) in GraphPad Prism would always give a *p* value >0.05 (https://www.graphpad.com/guides/prism/latest/statistics/how_the_mann-whitney_test_works.htm). After consultation with statistics experts in our Institute, we were advised to use *t*-test when comparing the means between two different groups.

5. For reproducibility of code and analyses, all analytical scripts for this study should be deposited in a repository such as GitHub or made available as a Supplementary file.

We appreciate this request and have provided the scripts as a Supplementary file (Appendix-Scripts). We refer the reader to this file in the Material and methods section.

Reviewer #2 (Recommendations for the authors):While we believe, the manuscript as a whole is close to publication quality, one of the major concerns is the reliance on cell lines as the major model system used, as opposed to primary cells and more in-depth in vivo analysis to strengthen the relevance of BAFF in aging. Please see our comments and concerns below:1) The amount of DNA damaging stimuli seems fairly weak (5gy IR and 10nM Doxo). Can the authors determine via cell cycle analysis or some other assay, what percent of cells are undergoing cell cycle arrest?

We thank the Reviewer for these questions and comments. Given that THP-1 cells are a leukemia line, they are much more sensitive to IR and Doxo than non-cancer cell types; therefore, we needed to choose doses of IR and Dox that would induce senescence without triggering significant cell death. In the revised manuscript (Figure 1—figure supplement 1B,C), we compared cell growth and viability of THP-1 cells after exposure to a single dose of 5 or 10 Gy. With 10 Gy, we observed a strong reduction in cell viability by day 6 after IR. Therefore, we decided to adopt 5 Gy for our experiments. To support the senescent state by 6 days after IR (5 Gy), we display several markers, including increased levels of p21 and DPP4 (Figure 1C), reduced BrdU incorporation and cell growth (Figure 1—figure supplement 1B,D), and increased SA-β-Gal and SPiDER-Gal activities (Figure 1A and Figure 1—figure supplement 1E). Finally, we analyzed the cell cycle distribution profiles of untreated and IR-treated THP-1 cells (Figure 1—figure supplement 1F). Under these conditions, we did not observe a clear cell cycle arrest, but rather a modest shift toward G2/M and S phase in irradiated cells. This response profile is in line with the current literature on senescent cancer cells; as reported in the Cancer SENEScopedia, senescent cancer cells lacking TP53 do not undergo growth arrest in a specific phase of the cell cycle (PMID 34320349).

Author response image 2 is the cell cycle distribution of THP-1 cells (6 days after treatment with IR or Doxo):

2) Likewise, in Figure 1 the SA-B-GAL staining in THP-1 cells seems to be only in a small subset of cells, can the authors also quantify the staining perhaps via the use of C12FDG using a plate reader or preferably flow cytometry?

We appreciate this valuable suggestion. In the revised manuscript, we have included quantitative SA-β-Gal analysis in IR-treated THP-1 cells (SPiDER-βGal, Dojindo), as reported in Figure 1—figure supplement 1E.

3) Figure 1: In addition to doxo-treated mice, can the authors also measure BAFF expression in old vs young tissues?

We thank the Reviewer for this question. We measured BAFF expression in old (27 months old, m.o.) versus young (3 m.o.) mice. To remain consistent with the Dox-treatment in Figure 1E-H, we analyzed spleen and serum, and used the same positive controls (*Cdkn1a* mRNA in spleen and secreted GDF15 for serum, respectively). We provide in Author response image 3 the data for the Reviewer. RT-qPCR analysis in spleen revealed that *TNFSF13B* mRNA levels are indeed slightly upregulated with aging. However, in serum, we observed a significant reduction of circulating BAFF with aging. More studies will be necessary to determine if this is due to a higher uptake of BAFF by recipient cells, or if instead BAFF is accumulated in local niches of senescent cells, rather than at the systemic level. We decided not to include these data in our manuscript because it will be important to address these questions in depth in future studies. Given that these mice were models of healthy aging, it will be interesting to study the role of BAFF in the context of specific age-related morbidities, in which senescent cells and inflammation play a crucial role.

**Author response image 3. sa2fig3:** 

4) Pg 6: "In all cases, BAFF mRNA levels increased during senescence (Figure 1—figure supplement 1E) and secreted BAFF was generally elevated with senescence, although it was undetectable in hVSMCs, and the levels were overall higher in THP-1 cells (Figure 1D and Figure 1—figure supplement 1F)." The total BAFF in the ELISA experiments in THP-1 and WI-38 cells is equivalent, in fact, the irradiated WI-38 cells have more expression than the irradiated THP-1 cells.

The Reviewer is quite right. We apologize for the confusion and have revised the text to describe the data more accurately.

5) Pg 6: "Next, we investigated if the rise in BAFF mRNA with senescence in THP-1 cells was the result of transcriptional or posttranscriptional regulatory mechanisms" Please explain the rationale for asking this question.

We thank the Reviewer for the question. Given our strong interest in gene regulation, particularly at the RNA level, we strive to identify the molecular factors that drive expression of key senescence-associated proteins. We have revised the sentence to be more descriptive:

“Next, to study the mechanism(s) underlying BAFF production in senescence, we investigated if the rise in TNFSF13B mRNA in irradiated THP-1 cells was the result of transcriptional or posttranscriptional regulatory processes”.

6) Figure 2G: The authors show that IRF1 and IRF2 are induced during senescence and may drive BAFF expression. Thus, in addition to IRF1 can the authors also silence IRF2 to test its effect on BAFF mRNA expression.

We thank the Reviewer for this interesting question. We silenced IRF2 in THP-1 cells following the same approach used for IRF1 silencing. However, we did not observe a reduction in *TNFSF13B* pre-mRNA, and instead we observed a minor increase. In Author response image 4 we provide the data for the Reviewer:

**Author response image 4. sa2fig4:** 

7) In Figures 4B and 4C, can the cell viability and cell proliferation be measured via flow cytometry which is more quantitative?

We thank the reviewer for this helpful suggestion. In the revised manuscript, we measured cell viability by flow cytometry, as advised, and cell numbers by XTT assay (Figure 4—figure supplement 1A,B). Details are in the revised Materials and methods and Figure legends.

8) Figure 5: To further define the BAFF signaling mechanism, can the authors also silence or KO the different BAFF receptors to determine which is the major receptor necessary for BAFF signaling in senescent THP-1 and fibroblast?

We thank the Reviewer for this interesting comment. We individually silenced each of the three BAFF receptors in THP-1 cells (Figure 5C and Figure 5—figure supplement 1A) and WI-38 cells (Figure 7—figure supplement 1C). In both cell types, silencing of *TNFSF13C* mRNA (encoding BAFFR) recapitulated part of the effects observed after BAFF silencing. Specifically, silencing BAFFR reduced the expression of some SASP factors in THP-1 cells, and reduced the levels of p53 in WI-38 cells. Interestingly, as described in the revised text, the silencing of BCMA or TACI had the opposite effect, suggesting that TACI and BCMA might be part of a negative feedback regulatory loop that regulates BAFF signaling. More studies are needed to confirm this hypothesis and to define the precise role of each BAFF receptor in senescence.

9) Figure 5: We suggest the authors perform an NFkB luciferase assay to confirm BAFF silencing impacts NFkB gene expression.

We followed the Reviewer’s helpful suggestion and co-transfected siRNAs and reporter constructs by electroporation in THP-1 cells. However, the co-transfection of plasmids and siRNAs was highly toxic, and we observed significant cell death especially after IR treatment. However, we were able to co-transfect a reporter plasmid [an *NF-ΚB*-driven promoter-reporter construct (pNF-κB-GFP) along with control reporters with a minimal promoter (pMin-GFP) or a constitutively active promoter (pCMV-GFP)] along with a BAFF expression plasmid (pBAFF) and an empty vector (pCTRL) by using lipofectamine LTX, a non-toxic reagent specifically indicated for transfection of DNA vectors. We performed this experiment in proliferating THP-1 cells, and we measured GFP signal in cells overexpressing BAFF (pBAFF) relative to control populations (pCTRL). This experiment is included in Figure 5-Supplemental Figure 1C.

10) Pg 12 Figure 6: "To gain a more complete understanding of the role of BAFF in senescence, we investigated its function in primary fibroblasts, which are well-established models for senescence and express the senescence-relevant protein p53" The subsequent experiments were not performed in primary fibroblast.

We thank the Reviewer for this comment. We neglected to explain that WI-38 and IMR-90 fibroblasts are primary cells. They were isolated from normal fetal lung tissue (WI-38 fibroblasts) or from the lung tissue of a 4-month old (IMR-90), both female, and have a finite lifespan in culture before they undergo senescence. We have indicated this in the revised manuscript.

11) Lastly, as mentioned above the paper would be strengthened if some of the key experiments can be performed in primary mouse or human monocytes and fibroblast to confirm BAFF regulates senescence in cells that are more relevant to normal physiology.

We thank the Reviewer for these suggestions. As explained above, we used primary WI-38 and IMR-90 fibroblasts derived from human lung tissue. Regarding monocytes, we obtained and tested primary CD14+ monocytes; however, their short lifespan in culture (2 to 4 days) and the inherent difficulty of transfecting them did not allow us to assess the role of BAFF in primary monocytes. Therefore, we used instead the two monocyte/macrophage-cell lines THP-1 (human, transformed) and RAW 264.7 (mouse immortal) macrophages. As our studies advance, we will develop more tools to enable the analysis of primary human monocytes/macrophages.

Reviewer #3 (Recommendations for the authors):1. Figure 1. Test whether the upregulation of BAFF is specific to senescence, or also in reversible quiescence arrest.

We appreciate the Reviewer’s requests. We performed the experiments in fibroblasts and THP-1 cells to assess BAFF levels in quiescence. As shown below in the figure for Reviewers, we induced quiescence in fibroblasts by serum starvation (0.1%) for 96 h and confirmed the quiescent state by measuring two markers of quiescence (reduction of *CCND1* mRNA and reduction of phopho-S6, when compared to cycling cells, following markers established previously [PMID 25483060]) (panel A). In this case, the level of *BAFF* mRNA was increased upon quiescence (panel B).

In THP-1 cells, we tried to induce quiescence by serum starvation and glutamine depletion for 96 h. Unfortunately, however, inducing quiescence in THP-1 cells was rather challenging, likely because they are cancer cells. Thus, we observed a reduction of cell proliferation in both conditions, but we observed a reduction in phospho-S6 only in the samples without glutamine (panel C). We failed to see increased *BAFF* mRNA levels in quiescent THP-1 cells after either serum starvation or glutamine depletion (panel D).

In summary, further studies will be necessary to fully understand if the increased expression of BAFF seen in senescent cells is also observed in other conditions of growth suppression (such as quiescence or differentiation), as well as whether this effect is specific to different cell types.

2. Figure 1, Supplement 1G. Show negative control IgG for immunofluorescence.

We thank the Reviewer for this suggestion. Along with other changes during the revision, we decided to remove the immunofluorescence data in order to include more informative data.

3. All results with siRNA should be validated with at least 2 individual siRNAs to eliminate the possibility of off-target effects.

We agree with the Reviewer on the importance of testing individual siRNAs. For BAFF, we originally tested two independent siRNAs (BAFF#1 and BAFF#2) individually, but we also pooled them for additional analysis (and referred to simply as “BAFFsi” along the manuscript). In the revised version of our manuscript, we included the key experiments performed with these two individual BAFF siRNAs. Upon BAFF silencing in THP-1 cells, we observed a reduction of SASP factors and SA-β-Gal activity levels with each individual siRNA (Figure 4—figure supplement 1D-F) and with the pooled siRNAs (Figure 4C). For WI-38 cells, we observed a reduction of p53 levels with individual and pooled siRNAs (Figure 7—figure supplement 1A), as well as a reduction in IL6 levels and SA-β-Gal activity (Figure 6—figure supplement 1D,E). After IRF1 silencing, we observed a reduction in *BAFF* pre-mRNA with two different pairs of CTRLsi and IRF1si pools (Figure 2I and supplementary Figure 2E). For the data on BAFF receptors, we used SMARTpools from Dharmacon, which are combinations of 4 siRNAs designed by the company to minimize off-target effects. These additions and clarifications are indicated in the revised manuscript.

4. To confirm a role for IRF1 in the activation of BAFF, the authors should confirm the binding of IRF1 to the BAFF promoter by ChIP or ChIP-seq.

We thank the Reviewer for this suggestion. We performed ChIP-qPCR analysis in THP-1 cells that were either proliferating or rendered senescent after exposure to IR (Figure 2H, Materials and methods section), and we confirmed the binding of IRF1 to the proximal promoter region of *TNFSF13B.* As anticipated, this interaction was stronger after inducing senescence.

5. Key antibodies should be validated by siRNA knockdown of their targets, for example, TACI, BCMA, and BAFF-R in Figure 5. Note that there is an apparent discrepancy between BCMA data in Figure 5B vs 5C.

We fully agree with the Reviewer on this point and we thank him/her for helping us to improve this part of our manuscript. To address the discrepancy regarding BCMA western blot analysis and flow cytometry data, we silenced BCMA in THP-1 cells and tested two different antibodies advertised to recognize BCMA. This experiment allowed us to identify the correct band for BCMA by western blot analysis. We then confirmed that BCMA is upregulated in senescence, as observed by both western blot and flow cytometry analyses. We have modified the manuscript to reflect these changes. Please find these data in Figure 5A,B and Figure 5—figure supplement 1A of the revised manuscript.

6. Figure 5E. Negative/specificity controls for this assay should be shown.

We thank the reviewer for this comment and regret that we were unable to provide a negative control. The kit only provides a competitive wild-type oligomer used to test the specificity of the binding. For each sample (CTRLsi, BAFFsi, CTRLsi IR, BAFFsi IR) and each antibody tested (p65, p50, p52, RelB and c-Rel), we evaluated the reductions in signal upon addition of excess competitive oligomer per well (20 pmol/well) compared to wells with an inactive oligomer. However, the negative control was performed only as single replicate, due to the limited quantity of nuclear extracts and the high number of samples and antibodies analyzed. We therefore considered this control as being ‘qualitative’ rather than fully ‘quantitative’.

7. Hybridization arrays such as Figure 5H, Figure 6 – Supplement 1I, and Figure 6H should be shown as quantitated, normalized data with statistics from replicates.

We appreciate this request. We have included the quantification and statistics to the phosphoarrays used for THP-1 and WI-38 cells, which had been performed in triplicate (Figure 7A, Figure 5—figure supplement 1D). The original arrays are shown in the respective Source Data Files. In the interest of space, we removed the cytokine array performed on IMR-90 cells and left instead the quantitative ELISA for IL6 (Figure 6—figure supplement 1F). The data obtained from the cytokine array analysis in Figure 4F and Figure 4-Supplemental Figure 1C are supported by quantitative multiplex ELISA measurements (Figure 4E and Figure 4C).

8. Figure 6B – Supplement 1. Controls to confirm fractionation (i.e., non-contamination by cytosolic and nuclear proteins) should be shown.

We thank the Reviewer for this suggestion. We tested the efficiency of fractionation and we did in fact observe some degree of contamination from cytosolic proteins using the earlier version of the kit (Pierce, cat. 89881). We therefore purchased an improved version of the kit (Pierce, cat. A44390) and repeated the surface fractionation assay, which this time showed improved fractionation (Figure 7—figure supplement 1B). Interestingly, with the improved fractionation strategy, we observed that BAFF receptors in fibroblasts were almost exclusively localized inside the cell and not on the surface, as we found in THP-1 cells. Further validation of BAFF receptor antibodies has been provided in Figure 5—figure supplement 1A. As described in the text, the intracellular localization of BAFF receptors was previously reported in other cell types and conditions (PMID 31137630, PMID 19258594, PMID 30333819, PMID 10903733), and thus it is possible that BAFF may act through non-canonical mechanisms in WI-38 cells. Nonetheless, we did detect a small amount of BAFFR on the cell surface, and furthermore, BAFFR silencing reduced the level of p53 in fibroblasts. Therefore, we propose that BAFFR may be the primary receptor involved in p53 regulation in fibroblasts (Figure 7—figure supplement 1B,C). Our data on BAFF receptors deserve deeper characterization in a future study of the functions of BAFF receptors in senescence.

9. Figure 6A. Knockdown of BAFF should be shown by western blot.

Yes, definitely. We appreciate this comment and have included BAFF knockdown data in fibroblasts by western blot analysis (Figure 7B).

10. Figure 6G. Although BAFF knockdown decreases the expression of p53, p21 increases. How do the authors explain this?

We thank the Reviewer for the interesting question. We too were surprised to observe that the p53-dependent transcripts regulated by BAFF did not include *CDKN1A* (*p21*) mRNA, as confirmed by western blot analysis. The accumulation of p21 in senescence can be also regulated by p53-independent pathways and in p53^-^/^-^ cells, for example by p90^RSK^, SP1, and ZNF84 (PMID 24136223, PMID 25051367, PMID 33925586). Eventually, we removed the data relative to p21 and γ-H2AX in favor of other data and to streamline the content of this manuscript for the reader.